# Personal Energy at Work: A Systematic Review

**Alexandra Francina Janneke Klijn \*, Maria Tims, Evgenia I. Lysova and Svetlana N. Khapova**

Department of Management and Organisation, School of Business and Economics, Vrije Universiteit Amsterdam, 1081 HV Amsterdam, The Netherlands; m.tims@vu.nl (M.T.); e.lysova@vu.nl (E.I.L.); s.n.khapova@vu.nl (S.N.K.)
\* Correspondence: afjklijn@gmail.com

**Abstract:** There has been a significant increase in studies on personal energy at work. Yet, research efforts are fragmented, given that scholars employ a diversity of related concepts. To bring clarity, we executed a two-fold systematic literature review. We crafted a definition of personal energy at work and a theoretical framework, outlining the dimensions, antecedents and boundary conditions. The theoretical implication of the framework is that it allows one to explain why—given similar work—some employees feel energized whereas others do not. The difference depends on the context that the employer offers, the personal characteristics of employees and the processes of strain and recovery. The paper concludes with a discussion of how future research can build on the proposed framework to advance the theoretical depth and empirical investigation of personal energy at work.

**Keywords:** energy; personal; work; vigor; thriving

## 1. Introduction

Research shows that energized employees can make a difference as they can better deal with change and experience less stress, which makes personal energy one of the success factors for improving business outcomes [1–5]. Because feeling energized at work is that impactful for performance and sustainable employability, Human Resource (HR) scholars and practitioners could benefit from knowing how to improve personal energy of employees. To achieve this, there needs to be an understanding of why some people feel energized under specific conditions and others do not. Thus, there needs to be a common ground on the meaning of personal energy at work.

However, there is a proliferation of applications and definitions of personal energy at work. Often researchers use different definitions of energy that are not explicit and are actually referring to the same concepts [6]. This construct proliferation withholds scholars from clear conceptual definitions that are essential for scientific progress [7]. Personal energy is referred to in research on vitality [8], vigor [9], work engagement [10,11], thriving [12,13], and recovery [14–16]. Similarities can be found, for example between the concepts of vigor and thriving where both concepts refer to experiencing a sense of vitality, feeling energetic and alert [9,12]. In addition, we found that energy can refer to the same concept but measure different things. For example, the construct of work engagement refers to vigor as one of the three dimensions. However, the concept of vigor as described by Shirom et al. [9] is not the same as in work engagement [17].

While researchers seem to agree that personal energy at work is important, we remain to have poor clarity of what this phenomenon entails. Especially the discussion regarding the dimensions of personal energy seems to raise disagreement among scholars. Literature on personal energy differs in describing the number of dimensions, ranging between one dimension [8], two dimensions [6], three dimensions [11,18–20] and four dimensions [14,21,22].

For example, Quinn et al. [6] state that personal energy is not mental, social, or spiritual as such, instead, people invest their physical energy in, or feel energized about, social, mental or spiritual activities. Kahn [11] explains that people invest physical, cognitive, and emotional energies in their work. Kahn's description of the dimensions of energy is

underlined by other scholars (e.g., Newton et al. [19]). Colet et al. [18] also suggest three energy dimensions: emotional arousal (e.g., feeling of enthusiasm), cognitive alertness (e.g., attention focus), and purposeful behavior (e.g., investment of physical resources). Three other dimensions of energy are discussed by Schippers and Hogenes [20]; mental energy (being able to intensely focus), physical energy (strength, endurance, flexibility) and emotional energy (being in touch with your feelings and core values). Fritz et al. [23] describe energy as having four different dimensions (physical, emotional, mental and spiritual) in relation to the work context and they urge scholars to examine what strategies are most potent for different energy dimensions. Pluta and Rudawska [21] discuss energies available by an employee as resources of employees consisting of four internal potentials: physical, intellectual, emotional and spiritual.

In addition, previous research on personal energy at work has been executed to further understand the construct. For example, Quinn et al. [6] conducted a thorough review related to personal energy in the work context and described how personal energy is related to constructs like flow, motivation, and resources. Kleine et al. [24] executed a meta-analysis of antecedents and outcomes of thriving at work. Pluta and Rudawska [21] conceptualized individual resources of employees based on the concepts of human capital and personal energy at work, and propose a framework of individual resources of employees that enables a holistic view of an individual in an organization. Baker [25] analyzed different levels of personal energy at work: micro (individual level emotional energy), meso (dyadic or relational energy), and macro level (group emotion, energy networks).

However, although some scholars executed thorough reviews on this topic, these papers either did not explore all dimensions of personal energy at work [18,24], did not systematically specify antecedents of personal energy [6], or did not execute a systematic review [21,25]. In short, efforts have been made to better understand the concept of personal energy at work, however research appeared to be fragmented. Whilst acknowledging the value of these earlier reviews, we argue that a new systematic review is needed that fully comprehends the construct of personal energy at work and in which a broad range of antecedents is included.

The focus of this article is on the clarification of the construct of personal energy at work, by identifying the core dimensions of personal energy at work and by creating an overview of its antecedents. Previous research has demonstrated the relationship between personal energy at work and business outcomes [1,2,4,5,26]. That being established, our aim is to identify how personal energy is formed. We believe that we need to know much more on the dimensions, the definition and antecedents to obtain clarity on the construct of personal energy at work.

To clarify the concept of personal energy at work and its definition, this study conducts a two-fold systematic literature review. This study is designed to address specific research questions by comprehensively collecting all the information available on the topic that is defined at the outset by absolute inclusion and exclusion criteria, following the PRISMA guidelines for systematic reviews, suggested by Liberati et al. [27]. Herewith contributing to the transparency, replicability and reliability of research regarding personal energy at work.

The review is guided by the following research questions:

RQ1: Which dimensions form personal energy at work?
RQ2: How is personal energy at work defined?
RQ3: What are the antecedents and boundary conditions of personal energy at work?

The contributions of this review are threefold. First, this review aims to provide clarity on the dimensions of personal energy at work, by offering a complete overview of the current literature. In order to advance knowledge about personal energy, our objective is to synthesize knowledge based on analyses of previous research on this topic that is derived from different fields (business, management and psychology) to achieve clarity on the dimensions and on how personal energy at work can be defined. Second, we contribute to a better understanding of personal energy at work by identifying the antecedents

and boundary conditions. Third, we synthesize this knowledge to deliver a theoretical framework of personal energy at work. This framework can serve as a basis to develop concrete recommendations for HR researchers and practitioners on actions they can take regarding personal energy at work.

The proposed theoretical framework, illustrated in Figure 1, shows how personal and contextual factors are related to personal energy at work, with the interference of the processes of strain and recovery. The framework proposes that personal energy at work consists of four dimensions: physical, emotional, mental and spiritual energy. The framework also shows the antecedents for personal energy at work; contextual factors (supervisor and environment) and personal factors (competence, behavior, psychological state and personality). Contextual factors also play a role in moderating the relationship between personal factors and personal energy at work. In addition, the processes of strain and recovery play a role in mediating between contextual/personal factors and personal energy at work. Finally, strain and recovery can moderate the relationship between personal factors and personal energy at work. We will further explain the foundation and justification of the theoretical framework in the findings section.

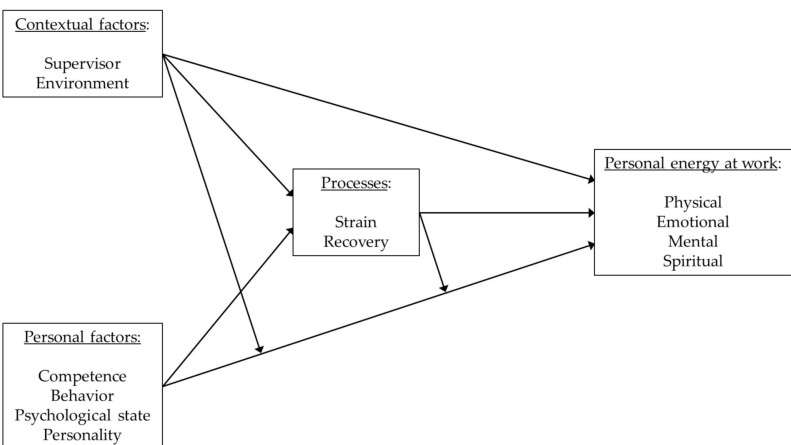

**Figure 1.** Theoretical framework for personal energy at work.

## 2. Theoretical Background

### 2.1. Personal Energy

Many concepts are referred to as, or relate to personal energy. For example, subjective vitality is defined by Ryan and Frederick [8] as a positive feeling of aliveness and energy that reflects people's wellbeing and covers both physical and psychological factors that impact the energy available to the self. The self-determination theory, described by Deci and Ryan [28], defines personal energy as subjective vitality or a sense of aliveness [8].

Another concept is vigor [9,16,29,30], described as the affective state that includes the feelings of physical strength, emotional energy, and cognitive liveliness [9,31].

Also thriving is a commonly used term [13,32–34]. The concept of thriving is clarified by Spreitzer et al. [13] in the socially-embedded model of thriving at work. Thriving represents the psychological state in which individuals experience both a sense of vitality and a sense of learning at work [13]. Thriving is seen as a dynamic state, a sense of progress or forward movement in the employees' self-development [35].

Quinn et al. [6]) describe a hierarchy of energy constructs and argue that human energy can be divided into physical energy and energetic activation. From their point of view, physical energy refers to chemical structures like glucose that are the drivers for kinetic energy, which enables individuals to move, do and think, and energetic activation refers to the degree to which people feel energized, which shares overlap with colloquial terms like emotional energy, mental energy and spiritual energy [6]. However, other researchers

argue that personal energy consists of four dimensions: physical, emotional, mental and spiritual energy [21–23].

Furthermore, the term collective or productive energy is mentioned in literature [18,36,37] and defined as the shared experience and demonstration of positive affect, cognitive arousal, and agentic behavior among unit members in their joint pursuit of organizationally salient objectives [18]. Although collective or productive energy might not be an individual-level construct, there are similarities in the dimensions of productive energy and personal energy at work. For example, Cole et al. [18] suggest that productive energy contains three domains: emotional arousal, cognitive alertness, and purposeful behavior like investment of physical resources. These domains are similar to three of the four dimensions mentioned by Fritz et al. [23]: emotional energy, mental energy and physical energy.

Concluding, researchers all describe the physical and psychological dimensions of personal energy at work [6,8,20,25,38], in which the psychological dimension is often subdivided into emotional, mental and spiritual energy [21–23].

### 2.2. The Dimensions of Personal Energy at Work

The description of personal energy as consisting of physical, emotional, mental and spiritual dimensions is a scheme found in similar forms in different fields of expertise (e.g., HR management, psychology, organizational behavior, business, and sports).

For example, Fritz et al. [23] propose that all four dimensions (physical, relational, mental, and spiritual) can be associated with positive energy experiences. In support of Fritz et al. [23] ), according to Pluta and Rudawska [21], personal resources of employees are the sum of four internal potentials that are at the disposal of physical, intellectual, emotional and spiritual energy. Loehr and Schwartz [39] argue that people need to tap into their energy at the four levels of performance with physical, emotional, mental and spiritual energy. Later on Schwartz further built on this theory and published practices for renewing physical, emotional, mental and spiritual energy [22]. Hrabe et al. [40] executed interventions to measure these four dimensions of energy on nurse practitioners and found, for example, a significant decrease in body fat percentage across time, based on tactics like healthy nutrition and exercise, that address the physical energy dimension. These four dimensions of personal energy at work are somewhat different from the description of Quinn et al. [6] who stated that energetic activation is an overarching dimension where people invest their physical energy in, or feel energized about, mental, social, or spiritual activities. Covey [41] mentions the four dimensions in his book 'The 8th habit', where he encompasses the four human intelligences in the description of human nature; each individual consists of the body, mind, emotions and spirit, and taking care of each of those dimensions contributes to better human activity. In the field of HR management, Dahlgaard-Park et al. [42] describe the muscle, heart, brain and spirit components that HR management practices focus on nowadays. Furthermore, physical, emotional, mental, and spiritual dimensions in regards to wellbeing are described by Graham and Martin [43] and a recent article on women athletes mentions that to flourish in sport, people need to be physically, mentally, emotionally and spiritually healthy [44]. According to Marques and Berry [45] the four dimensions physical, cognitive, social/spiritual, and psycho-emotional also refer to person centered resiliency.

Based on literature explaining personal energy at work [11,18,21,23,40], the approach of addressing the four dimensions of physical, emotional, mental and spiritual energy seems promising. The four interconnected dimensions make up for feeling energized where one dimension can be low while someone still experiences feeling energized. For instance, people can feel physically tired but still feel high spiritual energy due to fulfillment in work.

Concluding, there are many ways in which researchers have defined the construct of personal energy and four dimensions (physical, emotional, mental, and spiritual) are central to this construct. We choose to research the literature and investigate the dimensions of personal energy at work, to direct us in defining personal energy at work and to create clarity on the construct with its antecedents and boundary conditions.

### 3. Methods Review 1

In advancing construct clarity on personal energy at work we used different forms of knowledge synthesis, based on methods described by Locke and Golden-Biddle [46,47] and Post et al. [48], as these methods have been shown to offer proper guidance on systematic review methodology. This review was performed to identify the dimensions of personal energy at work. We used this synthesis to serve as a guide to identify how personal energy at work can be defined. This process is graphically depicted in the PRISMA [28] in Figure 2.

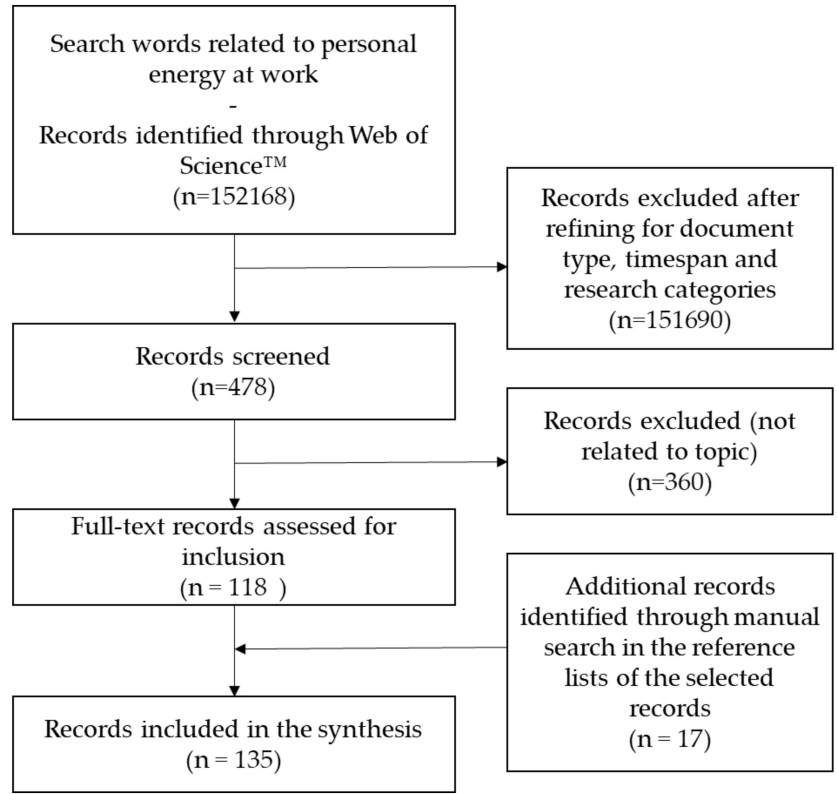

**Figure 2.** PRISMA review methodology process for personal energy at work, review 1.

We included articles from Web of Science™ because this database captures social sciences, it contains all academic disciplines, and it provides a sufficient stability of coverage to be used for more detailed cross-disciplinary comparisons [49]. An initial search by topic on Web of Science™ on 24 October 2021 resulted in 152,168 papers, after applying the following restrictions: Topic: (energy OR vitality) AND (work) AND (personal OR individual OR human). The results were then further refined by using the following options: document type: article and review and timespan: 2001–2021. Excluding all research categories except business, management, psychology, psychology applied and psychology multidisciplinary. After this, a total 478 of papers remained that were screened. We included papers that concerned the concept of energy related to a personal perception or experience of feeling energized. Main reasons for exclusion in this phase were records regarding employees in the energy industry or records on environmental energy use in companies. Papers were excluded when they lacked relevance to the topic and lacked academic thoroughness ($n = 360$). Finally, we conducted a manual search in the reference lists of the selected articles, which resulted in 17 additional references added to the list. As a result, the first synthesis is based on the full text analysis of 135 academic publications.

## 4. Findings Review 1

### 4.1. Construct of Personal Energy at Work

To obtain clarity on the construct of personal energy at work, we examined the 135 records to identify the dimensions of personal energy at work. This resulted in a taxonomy that formed the organizing structure for understanding previous research.

The first step was identifying the construct that was mentioned in each record. For example, in some papers vigor was researched as a construct on its own [29,30,49], other papers mentioned vigor as being part of the construct of work engagement [50–52]. The second step was exploring if the personal energy related construct was described as having different dimensions. For example, the construct of vigor can be measured on three dimensions: physical strength, emotional energy and cognitive liveliness.

A full overview of the 135 records with their constructs and dimensions is shown in Appendix A. The summary and count of these records with their constructs and dimensions is shown in Table 1. We found eight constructs used to refer to personal energy at work in the literature. Unidimensional energy constructs were found in 18 studies (e.g., Atwater and Carmeli [53]; de Bloom et al. [54]; Russo et al. [55]). In addition, 39 studies referred to energy as having one to four of the following dimensions: physical, emotional, mental, and spiritual energy (e.g., Loehr and Schwartz [56]; Schippers and Hogenes, [20]; Ten Brummelhuis et al. Productive energy, with three dimensions, was mentioned in three studies [18,36,37]. The construct of vigor was found in 28 studies, of which seven studies defined vigor using the dimensions of physical strength, emotional energy, and cognitive liveliness (e.g., Adil and Awais [29]; van Hooff and Geurts [57]). The other 21 studies mentioned vigor as a dimension of work engagement (e.g., Bakker [58]; Ozyilmaz [59]; Sanz-Vergel et al. [60]). The construct of thriving was mentioned in 22 studies, which all contained the thriving construct based on the dimensions of vitality and learning (e.g., Carmeli and Russo [33]; Cullen et al. [61]; van der Walt [34]). Vitality was mentioned in 25 studies, of which 17 studies were about vitality constructs (e.g., de Jonge and Peeters [2]; Op den Kamp et al [62] and eight studies concerned the construct of subjective vitality [63–65], all without dimensions.

**Table 1.** Count of constructs and dimensions of records included in review 1.

| Construct | *n* | Dimensions | *n* |
|---|---|---|---|
| Energy | 18 | - | - |
| Energy dimensions | 39 | Physical | 25 |
| | | Emotional | 23 |
| | | Mental | 23 |
| | | Spiritual | 9 |
| Productive energy | 3 | Affective | 3 |
| | | Cognitive | 3 |
| | | Behavioral | 3 |
| Vigor | 7 | Physical strength | 7 |
| | | Emotional energy | 7 |
| | | Cognitive liveliness | 7 |
| Vigor as part of engagement | 21 | - | - |
| Thriving | 22 | Vitality | 22 |
| | | Learning | 22 |
| Vitality | 17 | - | - |
| Subjective vitality | 8 | - | - |

In sum, it appeared that the energy construct with one to four dimensions of physical, emotional, mental, and spiritual energy is most used in previous research. Concluding, the results show that the four dimensions of physical, emotional, mental and spiritual energy were found in different combinations, and no additional dimension was identified.

Therefore, we propose that these four dimensions could serve as a broad basis for defining the construct of personal energy at work.

### 4.2. Dimensions of Personal Energy at Work

Synthesizing the knowledge on the four dimensions of personal energy at work, brings us to the following descriptions: Physical energy provides the strength for acting. It is the potential energy, obtained by nutrition, rest and sleep, that can be transformed into kinetic energy like movement [6,66]. Physical energy is experiencing physical strength, feeling vital and awake. Regular exercise with sufficient recovery can increase physical energy. Emotional energy enables the intention into motivation for action. It is about having positive feelings and experiencing acceptance and forgiveness [56]. Relationships play an important role in emotional energy. The presence of others and relationships can have a positive impact on feeling energized, for example when a manager shows empathy [67]. In addition relationships with colleagues can have a negative impact on personal energy, for instance when female-instigated incivility is present [68]. Mental energy enables selection of the efficient method of acting [21]. It is about being mindful, creative and having cognitive skills. Acting mindful refers to being mentally focused on a task or in a certain situation, experiencing a flow while working and taking conscious recovery breaks to restore energy [69,70]. Creativity refers to the production of novel, useful ideas or solutions [71]. Cognitive skills refer to learning and experiencing development [23,72]. Spiritual energy is the driving force for action in all dimensions of life [56], where the experience of purpose and meaningful work are key sources to stimulate this [3,73]. It is not linked to religion per se, for instance atheists can feel high in spiritual energy, as it concerns having spirit and finding fulfillment in each new day. Also a sense of calling [74] refers to spiritual energy. This is where people's core values lie. Core values refer to an individual's internal compass, as they make sense of all activities and are a source of guidelines for personal growth [41]. The key source that fuels spiritual energy is the courage and conviction to live by your core values.

Furthermore, we found that the dimensions are interconnected; they work together, relate, strengthen and diminish each other. To illustrate, a challenging physical workout can serve as a source of physical energy as well as lead to positive emotional renewal, which affects emotional energy [22]. Similarly, social interactions can be a source of both emotional and mental energy because positive social experiences lead to feeling emotionally energized and social interactions are a crucial factor in activating creativity among colleagues [71]. Spiritual energy is a unique force for action in all the four dimensions because it provides the source of direction [56].

However, the four energy dimensions can also diminish each other [56]. For example, when having low physical energy, like after having just a couple hours of sleep, or when having low blood sugar, people tend to get grumpy and short fused, which results in a lower level of emotional energy. In addition, lacking physical energy can lead to loss of concentration, which reflects mental energy. Another example, when experiencing strong negative emotions, like anger or feeling frightened, there is insufficient emotional energy, which can lead to loss of focus (mental energy) and lack of conscious feelings regarding living up to your core values (spiritual dimension).

### 4.3. Defining Personal Energy at Work

To contribute to greater construct clarity for personal energy at work, we followed the process suggested by Podsakoff [75] by executing the four stages for developing good conceptual definitions. In the first stage, we collected a representative set of definitions of personal energy at work by identifying all the related constructs. The different constructs and definitions are presented in Table 2.

**Table 2.** Overview of constructs related to personal energy at work.

| Author (Year) | Construct | Definition |
|---|---|---|
| Ryan and Frederick (1997) [8] | Subjective vitality | One's conscious experience of possessing energy and aliveness |
| Schaufeli et al. (2002) [17] | Work engagement | A positive, fulfilling, work-related state of mind that is characterized by vigor, dedication, and absorption |
| Loehr and Schwartz (2001) [39] | Energy | The capacity to engage in work |
| Shirom (2004) [9] | Vigor | The positive affective response to people's ongoing interactions with significant elements in their job and work environment that comprises the interconnected feelings of physical strength, emotional energy, and cognitive liveliness |
| Spreitzer et al. (2005) [13] | Thriving at work | The psychological state in which individuals experience both a sense of vitality and a sense of learning at work |
| Quinn and Dutton (2005) [76] | Energy | The feeling that one is eager to act and capable of acting |
| Cole et al. (2012) [18] | Productive energy | The shared experience and demonstration of positive affect, cognitive arousal, and agentic behavior among unit members in their joint pursuit of organizationally salient objectives |
| Quinn et al. (2012) [6] | Human energy in organizations | A dynamic concept that consist out of physical energy (stored as potential energy in the chemical bonds that make up glucose or adenosine triphosphate) and energetic activation (the subjective component of a biobehavioral system of activation, experienced as feelings of vitality, vigor, or enthusiasm) |

The second stage was the synthesis on the necessary, sufficient and shared attributes of the concepts related to personal energy at work, these are shown in Appendix A. In the third stage, we specified the dimensionality of the construct: the four most frequent dimensions in literature for constructs related to personal energy at work are physical, emotional, mental and spiritual energy. In stage four, we crafted a definition of the construct of personal energy at work.

That brings us to the following proposal for the definition of personal energy at work: Personal energy at work is an affective and dynamic state that is reflected through interconnected physical, emotional, mental and spiritual energy dimensions.

Similar as thriving [13], personal energy at work is a dynamic state, where employees feel more or less energized at any point in time. Employees can experience a range of energy experiences, rather than experiencing feeling energized or not. Depending on the circumstances and individual characteristics, a person can feel an energy increase, decrease,

or the same level of energy compared to a previous point in time. As the energy dimensions fluctuate, the feeling of personal energy at work fluctuates.

Furthermore, personal energy at work is a reflective construct, the dimensions depict the construct. This definition differs from other constructs like work engagement, vigor, vitality and thriving, as personal energy at work contains the physical, emotional, mental and spiritual dimensions, which are not or only partly represented in any other current construct definitions. In addition, we provide clarity about the construct being reflective, in contrast to the descriptions of previous constructs where no distinction was made between a reflective or formative construct [6,8,9,13,21,23].

### 4.4. Measuring Personal Energy at Work

We found that constructs with several specified dimensions are commonly measured with a validated scale that measures each distinctive dimension [9,12,17,72]. Therefore, we assume that measuring the multi-dimensionality of a construct could be more valuable than measuring an overall experience of feeling energized. This is underlined by the methodology used by Both-Nwabuwe et al [76,77]. Both-Nwabuwe et al. [77] identified instruments that measured meaningful work as a multidimensional construct. Both-Nwabuwe et al. [77] were interested in subscale reliability because the scales were developed on the basis of a priori multidimensional frameworks of meaningful work, therefore they suggest that each dimension should be measured by its own reliable subscale.

Based on our definition of personal energy at work, we set out to explore whether current measures that we found, might be able to assess all four dimensions of personal energy. In our search for constructs related to personal energy at work, we found several scales. The constructs of energy and vitality are commonly measured with questionnaires developed by Ryan and Fredrick [8] and Atwater and Carmeli [53]. These instruments assess an overall sense of feeling energized, without going into detail on any of the dimensions of feeling energized at work. Evaluating the scales that measured several dimensions, we found that the two scales of vigor and thriving, together measure all four dimensions.

Vigor represents the positive affective response to people's ongoing interactions with significant elements in their job and work environment that comprises the interconnected feelings of physical strength, emotional energy, and cognitive liveliness [9]. The vigor measure assesses the dimensions physical strength, emotional energy, and cognitive liveliness and thus covers three of the four dimensions of personal energy at work: five items that reflect the physical energy dimension (e.g., 'I feel I have physical strength'), four items on the emotional energy dimension (e.g., 'I feel capable of being sympathetic to coworkers and customers'), and five items on the mental energy dimension (e.g., 'I feel able to be creative'). We adopted the division of items proposed by Shirom [9], labeling the items for physical strength as physical energy and cognitive liveliness as mental energy. All items with elaboration on our rational where we build on the categorization proposed by Shirom [9], can be found in the Appendix B.

The thriving measure was developed by Porath et al. [12] based on the socially-embedded model of thriving at work from Spreitzer et al. [13]. Spreitzer et al. [13] stated that people cannot thrive without learning and development. However, without the vitality component accompanying the learning, cognitive development will be limited; beyond the tipping point of lacking vitality, cognitive gains will be lost (Spreitzer and Porath [4]). When developing the scale for thriving, Porath developed learning items and added these to the subjective vitality items already created by Ryan and Frederick [8]. Thriving plays a role in helping employees to achieve a form of self-adaptation that enables them to develop additional resources to feed the behaviors that promote thriving and enable their development Spreitzer et al. [13]. It is suggested that resources are renewable and produced through thriving at work, where four types of resources can be generated: knowledge, positive meaning, positive affective resources and relational resources Spreitzer et al. [13]. This implicates that finding meaning, spiritual energy, is part of thriving. In addition, we found that Fritz et al. [23] used the same measurements for vitality as Porath used for

thriving which were derived from Ryan and Frederick [8], to examine physical, relational (which is linked to emotional energy), mental, and spiritual energy.

After analyzing the scale for thriving developed by Porath et al. [12], we conclude that the thriving scale covers three of the four dimensions of personal energy at work: three items on the physical dimension, five items on the mental dimension, and two items on the spiritual dimension. Although the two dimensions of thriving are labeled learning and vitality, we consider the learning dimensions also to be a form of energy, namely mental energy. We consider all items regarding learning to be part of the mental dimension (e.g., 'I find myself learning often'), as the mental dimension concerns cognitive skills, where learning and development are key [23]. To our opinion the items regarding the vitality subscale of thriving address the physical as well as the spiritual dimension. Items referring to the physical dimension concern the experience of having physically enough energy (e.g., 'I feel alive and vital'). Items from the vitality subscale of thriving that are referring to the spiritual dimension address having spirit and finding purpose in each new day (e.g., 'I have energy and spirit'). We elaborate our rationale in the items list in Appendix B.

Summarizing, there is no validated scale available that measures physical, emotional, mental and spiritual energy together. However, the scales of Shirom et al. [9] for vigor and Porath et al. [12] for thriving, together address the four dimensions. Where both scales include items on the physical and mental dimension that are complementary, emotional energy is represented in the vigor scale and spiritual energy is represented in the thriving scale.

Although personal energy at work is not operationalized within the constructs of vigor or thriving, reviewing the studies about vigor and thriving can provide knowledge on what factors lead to the experience of personal energy at work. Therefore, we chose to analyze literature regarding vigor and thriving in review 2 to identify the antecedents and boundary conditions of personal energy at work.

## 5. Methods Review 2

We synthesized knowledge of research papers that identified the antecedents and boundary conditions of vigor or thriving, because the instruments of vigor and thriving together address the physical, emotional, mental and spiritual dimension of personal energy at work.

We analyzed literature that researched vigor or thriving on the level of each distinct dimension. We excluded literature that measured the whole construct of vigor or thriving and therefore we included only quantitative papers as they addressed the individual dimensions. We selected the vigor papers based on their use of the Shirom's scale [9] and the thriving papers all contained measurements of the scale developed by Porath et al. [12].

We included empirical research that used validated or adapted instruments that measured vigor or thriving as a multidimensional construct. This process is graphically depicted in the PRISMA [27] in Figure 3. For vigor we conducted a literature search on Web of Science™ on 25 October 2021 on all papers that referenced the study of Shirom et al. [9], in which the definition of vigor and its measurement scale of vigor was developed. This search resulted in 152 papers being screened, of which 118 were excluded because they did not include the scale measurement developed by Shirom et al. [9] and antecedents of vigor, resulting in 34 studies that were included in the review. These 34 studies regarding vigor are also described in Appendix C.

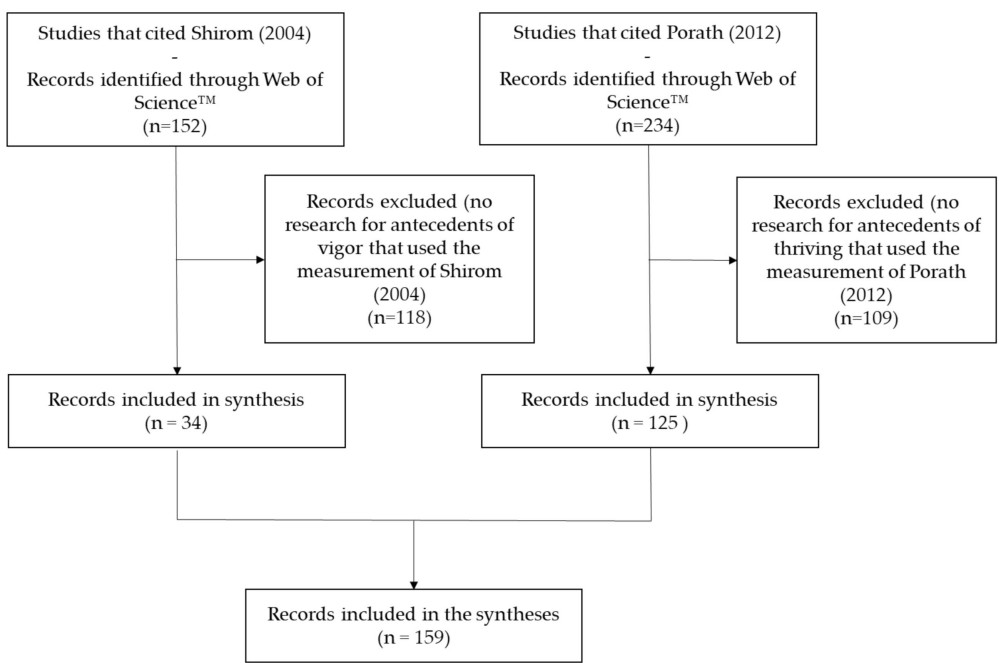

**Figure 3.** PRISMA review methodology process for personal energy at work, review 2.

For thriving we conducted a literature search on Web of Science™ on October 25, 2021 on all papers that referenced the study by Porath et al. [12] in which the measurement scale of thriving was established. This search resulted in 234 papers, of which 109 were excluded because they did not include the measurement developed by Porath et al. [12] and antecedents of thriving, resulting in 125 studies that were included in the review. These 125 studies concerning thriving are also presented in Appendix C.

## 6. Findings Review 2

### 6.1. Antecedents of Personal Energy at Work

We found that the antecedents can be broadly divided into two groups: personal factors and contextual factors. Personal factors are the personal resources, characteristics and actions of employees.

For example, Experiencing stress is a physical and psychological process activated by hormones, existing within the individual who is experiencing it [15]. This means that feeling energized at work is also about what an individual perceives as an energy driver or energy inhibitor and how to act on it. In addition, contextual variables are resources that the employer provides, such as flexible working hours and demands like targets that need to be achieved [10,58]. This implies that there is a dual responsibility for increasing personal energy at work: for the employer as well as the employee.

We categorized our findings in Appendix C, where we summed all antecedents and moderators that have shown to impact vigor or thriving. A summary of our finding regarding the antecedents of vigor and thriving is presented in Table 3. We discovered six similar themes of antecedents for vigor and thriving: the personal antecedents being competence, behavior, psychological state and personality, and contextual antecedents being the supervisor and the environment. The personal and contextual factors are similar to the antecedents mentioned in previous review analyses, like individual characteristics and relation resources, mentioned in a meta-analysis of thriving at work [24]. However, we believe that our terminology is more complete than Kleine et al. [24] as we go beyond individual characteristics and relationships by including environmental variables. We will explain this and go into detail on each of the antecedent themes.

**Table 3.** Antecedents of vigor and thriving.

| Antecedent Theme | Antecedents of Vigor | Antecedents of Thriving |
|---|---|---|
| Competence | Coping (+)<br>Job control (+)<br>Performance (+)<br>Customer orientation (+) | Mental toughness (+)<br>Goal progress (+)<br>Self-efficacy (+)<br>Optimization (+)<br>Political skill (+)<br>Problem solving (+)<br>Strength use (+)<br>Career adaptability (+)<br>Social functioning (+)<br>Overqualification (−)<br>Goals that match academic context (+)<br>Feedback-seeking from team members (+)<br>Cultural intelligence (+) |
| Behavior | Physical exercise (+)<br>Health (+)<br>Task focus (+)<br>Heedful relating (+)<br>Exploration (+)<br>Positive perfectionism (+)<br>Off-job time spent on social activities or low effort activities (+)<br>Proactive vitality management (+)<br>Job crafting (+)<br>Negative perfectionism (−)<br>Surface acting (−)<br>Off-job time spent on work-related activities (−)<br>Recovery activities (+) | Heedful relating (+)<br>Task focus (+)<br>Promotion focus (+)<br>Prevention focus (−)<br>Mindfulness practice (+)<br>Learning goal orientation (+)<br><br>Exploration at work (+)<br><br>Counterproductive work behaviors (−)<br>Customer orientation (+)<br>Self-attributed goals (+)<br>Online leisure crafting (+)<br><br>Speaking out behavior (+)<br><br>Helping coworkers (+)<br>Taking charge (+)<br>Job crafting (+) |
| Psychological state | Emotional stability (+)<br>Bonding social capital (+)<br><br>Secure attachment (+)<br>Detachment from work (+)<br>Job satisfaction (+)<br>Engagement (+)<br>Job involvement (+)<br>Satisfaction of need for relatedness (+)<br>Positive work reflections (+)<br><br>Self-verification striving (+)<br><br>Counterdependence (−)<br>Overdependence (−)<br>Burn out (−)<br>Negative work reflections (−) | Believing in incremental theories (+)<br>Self-assurance (+)<br>Fairness perception (+)<br>Job satisfaction (+)<br>Prosocial motivation (+)<br>Relational attachment (+)<br>Psychological detachment (+)<br>Affective rumination (−)<br><br>Positive work reflections (+)<br><br>Negative work reflections (−)<br>Paradox mindset (feeling positive about tensions) (+)<br>Hope regarding future work (+)<br>Psychological empowerment (+)<br>Self-esteem (+)<br>Autonomy satisfaction (+)<br>Work meaningfulness (+)<br>Psychological capital (+)<br>Psychological availability (+)<br>Having a calling (+)<br>Perceived challenge stressors (+)<br>Perceived hindrance stressors (−)<br>Psychological well-being (+)<br>Reattachment to work (+)<br>Activated positive affect (+)<br>Intrinsic motivation (+)<br>Perceived employability (+)<br>Harmonious passion (+)<br>Obsessive passion (+) |

**Table 3.** *Cont.*

| Antecedent Theme | Antecedents of Vigor | Antecedents of Thriving |
|---|---|---|
| Personality | Extraversion (+)<br>Agreeableness (+)<br>Openness (+)<br>Consciousness (+) | Feeling mindful (+)<br>Focus on opportunities (+)<br>Occupational future time perspective (+)<br>Interpersonal justice (feeling fair about company methods) (+)<br>Perceived insider status (feeling of belonging) (+)<br>Eudaimonic motives (+)<br>Hedonic motives (+)<br>Basic psychological need satisfaction (+)<br>Positive affect (+)<br>Feeling performance pressure (+)<br>Extraversion (+)<br>Consciousness (+)<br>Neurotism (−)<br>Proactive personality (+)<br>Personality trait of core self-evaluations (+)<br>Personality trait: hope (+)<br>Personality trait: optimism (+) |
| Supervisor | Leader–member exchange (+)<br>Supervisor support (+)<br>Supervisory working alliance (+)<br>Leaders' energy (+) | Empowering leadership (+)<br>Authentic leadership (+)<br>Managers' empathy (+)<br>Challenge/positive appraisal (+)<br>Hindrance/negative appraisal (−)<br>Directive performance management (−)<br>Enabling performance management (+)<br>Controlling style of coach (−)<br>Supervisor support (+)<br>Transformational leadership (+)<br>Leader–member exchange (+)<br>Managerial coaching (+)<br>Role overload (−)<br>Role ambiguity (−)<br>Servant leadership (+)<br>Negative appraisal (−)<br>Positive appraisal (+)<br>Role ambiguity (−)<br>Inclusive leadership (+)<br>Provide task identity (+)<br>Provide autonomy (+)<br>Mentoring (+)<br>Ambidextrous leadership (+)<br>Family supportive supervisor behaviors (+)<br>Zhongyong leadership (integrating the opinions of all parties and pursuing harmony) (+)<br>Participative leadership (+)<br>Paradoxical (fair) leader behavior (+)<br>Differential leadership (+)<br>Abusive supervision (−) |
| | Job insecurity (−)<br>Mindfulness intervention (+)<br>Social capital (+)<br>Work–home conflict (−)<br>Collective set of pressure, threat, financial strain, relationship conflict and supervisory working (−)<br>General work related threats (−)<br>Organizational support (+) | Job deprivation on cultural instruction competence (−)<br>External challenge stressors (−)<br>External hindrance stressors (+)<br>Employee involvement climate (+)<br>Perceived organizational support (+)<br>Psychological safety (+)<br>Workplace civility (+) |

**Table 3.** *Cont.*

| Antecedent Theme | Antecedents of Vigor | Antecedents of Thriving |
|---|---|---|
| | Presence of a garden (+) | Control governance practices (+) |
| | Second shift workload (−) | Collegial governance practices (+) |
| | Social support (+) | Fairness/justice (+) |
| | Sports demands (−) | Human resource practices (+) |
| | Cognitive resources (+) | Female-instigated incivility (−) |
| | Cognitive demands (+) | Workplace incivility/violence (−) |
| | Emotional demands (−) | Person centered climate (+) |
| | Emotional resources (+) | Perceived organizational support for strength use (+) |
| | Physical demands (+) | Mutual understanding (+) |
| | Employment security (+) | Reciprocal favor (+) |
| | Perceived morning weather (−) | Relationship harmony (+) |
| | | High-performance work system (+) |
| | | Colleagues show compassion/civility (+) |
| | | Lean maturity (+) |
| | | Job resources (+) |
| | | Quality of intergroup contact (+) |
| | | Team innovation climate (+) |
| | | Team empowerment climate (+) |
| | | Intrapersonal identity conflict among colleagues (−) |
| | | Psychological contract fulfillment (+) |
| | | International business travel with high responsibility for people (+) |
| | | International business travel with low responsibility for people (−) |
| | | Co-worker support (+) |
| | | Strong relational ties with colleagues (+) |
| | | Weak relational ties with colleagues (+) |
| | | Training and development (+) |
| | | Mindfulness intervention provided (+) |
| | | Citizenship fatigue (−) |
| | | Sense of community (+) |
| | | Experienced service-oriented high-performance work systems (+) |
| | | Team-member exchange (+) |
| | | Positive career shocks (+) |
| | | Problems at home (−) |
| | | Needs-supplies fit (+) |
| | | Work-to-family conflict (−) |
| | | Family–work enrichment (+) |
| | | Job complexity (+) |
| | | Corporate social responsibility (+) |
| | | Work demands hindrance stressors (−) |
| | | Work resources challenge stressors (+) |
| | | Decent work (eg. treated with dignity) (+) |
| | | Perceived competitive climate (+) |

Our classification is based on the setting and quantitative research outcomes mentioned in the articles. Meaning, we allocated a variable to be a personal or contextual antecedent when the article positioned it this way. For example, job crafting (as an agentic work behavior) is described by Guan and Frenkel [78] as employees seeking resources by securing information and feedback from supervisors and colleagues and searching for learning opportunities. Therefore, we allocated job crafting to the personal behavior theme. The allocation of an antecedent to be positive (+) or negative (-) refers to the quantitative research outcomes of the articles where positive or negative relationships were identified between the independent variable and vigor or thriving.

## 6.2. Personal Antecedents

We identified four themes of personal factors that are found to correlate with personal energy: *competence* (what I can), *behavior* (what I do), *psychological state* (how I feel) and *personality* (who I am).

*Competence* refers to the personal antecedents that are characteristics or skills that make it possible to perform a task. We found evidence in studies that competence is positively related to personal energy at work. For example, a competence like political skill, explained as the ability to understand social and political aspects in the workplace and use that understanding to effectively influence others, has been found to enhance workplace thriving [61]

*Behavior* is a function of the decisions of actions that individuals make in conjunction with themselves or their environment [79]. Spreitzer et al. [13] stated that task focus, exploration and heedful relating are individual agentic work behaviors that drive thriving at work. These variables were confirmed by other studies to impact personal energy at work [12,80,81]. When employees act in an agentic way, they produce resources like positive affect and knowledge for themselves [82]. Paterson et al. [81] refined this as those who are more active and purposeful at work are more likely to experience and sustain vitality and learning. Task focus occurs when an employee is attentive and alert during the performance of work-related tasks and fully engaged in the task at hand by voluntarily and intentionally driving their personal energy into the work task [11].

The *psychological state* is the theme that contains constructs like emotions, beliefs, motivations and satisfaction. An example of a negative relationship with vigor is counter-dependence, where employees are more likely to avoid dependence on colleagues based on their belief that others will not be there for them in times of need, and this negatively relates to feeling vigorous [83]. A psychological state that is positively related to personal energy at work is relational attachment. Ehrhardt et al. [84] examined relational attachment, defined as the cumulative experience of feeling connected, attached, and close to others at work. These scholars found that employees become more attached to their organization and experience more beneficial outcomes like feeling energized, when they have stronger psychological attachment to colleagues at work.

*Personality* is the quality or characteristic that distinguishes the character and attitude of a person. Neuroticism is one of the big five personality traits that has a negative relationship with thriving [85] and inhibits personal energy at work. Openness is positively related to personal energy at work, as openness positively relates to the vigor component in work engagement [86].

## 6.3. Contextual Antecedents

We identified two contextual themes that impact personal energy at work: *the supervisor* and *the environment*. We categorized the supervisor as a different antecedent theme than the environment because environmental factors concern work conditions like a psychological safe environment, workplace violence, interventions and second shift workload. The supervisor factors concern the personal relationship between the supervisor and the employee, like managerial coaching and the managers' empathy. Therefore, we position the supervisor as an extra influencing variable on top of the environmental factors.

The *supervisor* or line-manager is a person with direct managerial responsibility for one or more employees. The importance of a line-manager is underlined by the Gallup studies: the line-manager accounts for at least 70% of the variance in employee engagement scores across business units [87]. The supervisor seems to have a significant impact on personal energy at work. An example of the supervisor is being positively related to vigor is leader-member exchange. A good in-group relationship between the member and the leader (leader-member exchange) is positively associated with the individual's feeling of energy [29]. Supervisors can be negatively related to personal energy at work when they cause role overload or role ambiguity [61]. It is the supervisors' responsibility to make

sure that employees know what is expected of them for employees to feel more energized at work.

Like the supervisor, the *environment* is also a contextual antecedent theme that can be a positively as well as negatively related to personal energy at work. Examples of the environment being positively related to personal energy at work are psychological safety [88], lifestyle intervention offerings like mindfulness trainings [89] and a climate that stimulates involvement of employees [90]. Employee involvement climate means that employees can make decisions and act on them, access and share the informational resources needed, have opportunities to learn and are rewarded, and this positively relates to thriving [90].

To summarize, based on the systematic literature review, six antecedent themes for personal energy at work were identified; competence, behavior, psychological state, and personality that are categorized as personal factors, and the supervisor and the environment represent contextual factors. This overview of antecedents provides us with more clarity on the factors that influence personal energy at work. Nevertheless, we need to understand how employees can cope with the inhibiting antecedents, for companies to facilitate in enhancement of personal energy at work. Therefore, we explored the moderating and mediating factors.

### 6.4. Factors That Moderate and Mediate Personal Energy at Work

Personal energy at work is influenced by a complex combination of factors, going from antecedents directly to the dimensions of personal energy at work, might be incomplete as there are other factors that influence this relationship.

Factors that moderated the relationship between other variables and vigor or thriving were also measured in the literature that we identified in review 2. We found 44 moderators, the full list of the moderators can be found in Appendix C. There were 24 moderators labeled as contextual factors. Environmental factors like employee involvement climate positively moderate the relationship between personal behavior and thriving, such that this relationship becomes stronger as employee involvement climate is higher [90,91]. Another moderator was mentoring of the manager. Mentoring moderates the effects of task identity and autonomy on thriving at work, such that these effects will be stronger for those who receive lower-quality mentoring [92].

More examples of moderators were found in factors like strain and recovery [22,26,93–95]. Personal strain can be described as severe or excessive demand on the strength, resources, or abilities of someone [96]. Strain or stress is often perceived as a negative influencer, but it can also serve as an enhancer of energy, as long as there is sufficient recovery [97].

Recovery is described as the process that replenishes physical and psychological resources that are essential for human functioning and wellbeing [98]. Recovery can be seen as a process opposite to strain, that results in restoration of impaired health [16]. Recovery can be obtained by psychological, mental and physical relaxation, that can be achieved during work [23,99,100] and after work [14,16,101]. Recovery during work examples are doing mediation and breathing exercises. Companies can stimulate recovery, for example by providing proper time for lunch breaks, without lunch meetings being scheduled.

Strain and recovery can have an impact on personal factors that are related to feeling energized [6,15,21,30,49,93,102–104]. Techniques to switch between strain and recovery have shown to be of impact on all four energy dimensions, where strain is the stimulus to growth, and recovery is where growth happens [56]. Some argue that growth only occurs in response to strain [97]. For example, to increase physical energy with exercise people need to grow their muscles, development occurs when a muscle is systematically worked beyond its capacity. Without that strain, there is no growth and development in the strength of the muscle. Strain can serve as a catalyst for physical changes, however it can advance a physical state toward either health or disease [105]. Muscle development also requires nourishment and time to consolidate and recover, without time to rest, exercise

is counterproductive [97]. Like physical growth, strain can also induce the enhancement of the other dimensions of personal energy at work, like growth on emotional, mental or spiritual level. For example, with regards to the mental dimension, cognitive skills decrease in the absence of intellectual challenge. However, having too many cognitive demanding tasks will work contradictory.

We discovered that processes like strain and recovery link to different antecedent themes, depending on how it is measured. For example, feeling stressed is a psychological state within the person [106], and stress is having physical and psychological consequences when it is manifested as burn out [93]. In addition, an external hindrance stressor can cause people to feel strained, for example when employees are been withheld from further development [107]. Similarly, recovery can originate within the person, as well as triggered by the environment. Recovery can be a behavior when doing mindfulness practices [108,109] or recovery during work can be initiated by the employer, for example by offering mindfulness interventions [89,110]. From a medical point of view, when activated by internal or external triggers, psychological strain can arise caused by hormonal chemical activation, having psychological and physical consequences [111]. Similarly, recovery can be activated internally or externally. Concluding, we propose that both strain and recovery can serve as additional factors that influence personal energy at work.

Examples of strain associated moderators that influence personal energy at work are burn out and emotional exhaustion. Time spent on work-related activities during off-job time has a stronger negative relationship with vigor for employees high (versus low) in burn out [93]. High emotional exhaustion negatively moderates the relationship between perceived transformational leadership and thriving, and low emotional exhaustion positively moderates this relationship [26]. The ability to be full on, full off at work and at home is impacted by how employees deal with stressors and mentally recover throughout the week [22]. This is in line with the research of Cheng at al. [95] where they investigated the extent to which employees think about non-work issues while being at work. They discovered that mentally segmentation between home and work, like employees who 'turn off' family stressors during work, moderate the relationship between problems at home and thriving, such that the relationship is weaker when employees have high rather than low segmentation [95]. Another example of recovery being a moderator is found in research from [94], where they discovered that recovery moderates the relationships between running-related demands and mental/emotional energy, e.g. runners facing high physical demands reported higher cognitive liveliness when they had high physical recovery.

We conclude that both strain and recovery processes can act as moderators influencing personal energy at work and the personal factors; competence (what I can), behavior (what I do), psychological state (how I feel) and personality (who I am) [22,26,93–95].

Because personal characteristics and external factors can initiate strain and recovery, and impact the dimensions of personal energy at work, we argue that the processes of strain and recovery can also serve as mediators between the antecedents and the dimensions of personal energy at work. The Conservation of Resources (COR) theory [103] supports this assumption. To explain differences in the ways in which people handle stressors, the COR theory was developed [103]. In the COR theory, resources are defined as things that people value and the experience of a resource loss, or the threat of losing resources, causes feelings of stress and in reaction people attempt to retain and protect their resources ([112]. Hobfoll [103] describes that stress acts as a mediator for people's conditions, conditions in this context are described as resources to the extent that they are valued and sought after.

This theory is underlined by scholars who suggest that handling strain or recovery drives the relationships between personal/contextual factors and feeling energized at work [16,21–23]. In addition, Lin et a. [106] discovered that challenge stressors positively mediate the role of the transformational leadership in increasing employees' thriving at work.

Summarizing, we found evidence for four moderator themes: supervisor, environment, strain and recovery. Literature showed that contextual factors (environment and the

supervisor) act as moderators between personal antecedents and personal energy at work. In addition, the processes of strain and recovery can moderate the relationship between personal antecedents and personal energy at work. Furthermore, we propose that strain and recovery can serve as a mediator between personal/contextual factors and the personal energy at work.

Concluding, employees' experiences of feeling energized is shaped by these antecedents and boundary conditions. Meaning that each dimension of personal energy at work is affected by contextual factors (supervisor and environment) and personal factors (competence, behavior, psychological state and personality), and the processes of strain and recovery.

Derived from the knowledge in the reviewed literature and the resulting analyses, a theoretical framework for personal energy at work emerges as a result. Figure 1 demonstrates the theoretical framework of personal energy at work, where the relationships are presented among the antecedents, boundary conditions and the construct of personal energy at work.

## 7. General Discussion

The purpose of this paper was to provide construct clarity of personal energy at work by executing a two-fold systematic literature review. We have come to three mayor findings. First, personal energy at work consist of four dimensions; physical, emotional, mental and spiritual energy. We crafted the definition of personal energy at work: Personal energy at work is an affective and dynamic state that is reflected through interconnected physical, emotional, mental and spiritual energy dimensions.

The second main finding was that personal energy at work can be measured by addressing the four energy dimensions. Presently, there is no validated scale available that measures all dimensions of physical, emotional, mental and spiritual energy at once, however, the scales for vigor [9] and thriving [12] together, cover these dimensions. Because the measurement scale for vigor [9] and the scale for thriving [12] address the four energy dimensions, we reviewed literature regarding antecedents and moderators that have been shown to impact the dimensions of vigor and thriving.

The third finding is that we identified all antecedents and moderators of vigor and thriving, and then extrapolated these to antecedents and boundary conditions of personal energy at work. Personal and contextual factors are related to feeling energized in each of the dimensions. Contextual factors moderate the relationship between the personal factors and personal energy at work. The processes of strain and recovery moderate the relationship between personal factors and personal energy at work. In addition, the processes of strain and recovery mediate the relationship between personal/contextual factors and the construct personal energy at work. This led to the proposal of the integrated theoretical framework of personal energy at work that can be a valuable instrument for researchers and practitioners in the field of HR management.

### 7.1. Theoretical Implications

Our primary theoretical contribution is that we synthesized knowledge and provided clarity about how personal energy is a reflective construct where the four dimensions are interconnected. This allows scholars to understand the phenomena better than before, where previously the knowledge about personal energy at work was scattered.

To our knowledge, no existing reviews to date have explored the antecedents and boundary conditions based on the construct dimensions. We offer contributions to the field of energy dimensions [6,21,41,42,56], overall feeling of energy [2,29], vigor [9,30], thriving [12,13,24], strain [113,114] and recovery [14,115], in a way that we combined knowledge from all these fields and we synthesized this knowledge that resulted in crafting the definition and the development of the theoretical framework of personal energy at work.

The theoretical framework of personal energy at work is a distinctive framework because it is the first framework to present the dimensions of personal energy at work and to categorize the factors that lead to the experience of feeling energized at work. Earlier research did not specify the antecedents addressing all four dimensions [6,9,13,14,24]. The framework explains why some employees feel energized compared to those who do not, given similar work. The difference depends on the context that the employer offers, the personal characteristics of employees, and how employees use processes like strain and recovery to enhance their personal factors.

The theoretical framework of personal energy at work can prevent further fragmentation of energy theories over different fields. The theoretical framework can serve as a starting point for researchers interested in further exploring personal energy at work in a consistent way. The construct clarity on personal energy at work is critical for building new theory, like the four distinct dimensions that are impacted by different personal and conceptual factors. In addition, construct clarity is a necessity for developing measurements.

We build on recent efforts to acknowledge theory of four energy dimensions [6,22,23] and altered the contribution of personal and contextual factors by addressing each dimension with the interference of strain and recovery processes. Our theoretical framework serves to create awareness that feeling energized is more than just getting enough sleep; it is physical, emotional, mental and spiritual.

Personal energy at work grows by enhancing the four dimensions, the way to do that is addressing all variables that influence these dimensions: personal factors, contextual factors and the processes of strain and recovery. We should adapt the distinction that previous measurements have made between different dimensions of the construct, however what is lacking is a complete referral to al dimensions: physical, emotional, mental and spiritual. For example, previous research like Kleine et al. [24] did not clearly distinguish the antecedents leading up to each energy dimension. Scholars should add the dimensions and take into account that the processes of strain and recovery are essential variables that can influence personal energy at work.

This paper provides an overview that supports HR researchers and practitioners in making connections between the various theories and frameworks that have been used to study personal energy. By observing the importance of vigor as well as thriving to be of absolute prominence in helping employees to improve their personal energy at work. What we should retain from existing measurement methods are the validated scales for the constructs as stated: vigor and thriving. However, we must realize that these constructs are separately not fully covering personal energy in the work context.

Our contribution comes from our articulation of a set of theoretical mechanisms that explains how employees and employers can enable personal energy at work. Although it is widely recognized that individual experiences in organizations are jointly shaped by characteristics of employees, supervisors and the work environment, limited research in HR recognizes both factors as important co-determinants of personal energy at work [21,116].

A theoretical implication is that employees who manage to wisely use their competence (what I can), behavior (what I do), psychological state (how I feel) and personality (who I am), are more likely to feel more energized. In addition, employers who provide proper energy management and a healthy environment are likely to enhance personal energy at work. Our framework contributes to the awareness that employees themselves have four factors that influence personal energy at work: their competence, behavior, psychological state and personality. In addition, employers have two influencers in hand to support employees: the supervisor and the environment, and that these two factors can also induce strain and recovery.

The antecedents identified for personal energy at work can serve as terms necessary to obtain energy. In addition, strain and recovery could be perceived as key indicators to achieve an energized workforce. Individuals are not simply influenced by their contexts, but also the process of recovery helps to renew their resources in performing their job [114], and in this way they can actively shape their personal contribution, even more so when

the employer supports recovery. Furthermore, stressors, mostly activated by external factors [113] can be dealt with by developing coping mechanisms [117].

Reflecting on how personal energy at work has developed and is embedded in organizational behavior, we can make assumptions for the future. For example, now in 2021 we live in a worldwide pandemic, the focus on employee health and wellbeing has never been higher [118]. Working at home with flexible working hours has become the new norm [119] where some employees flourish due to the new working conditions and others experience more stress or even burn down. Taking these personal and environmental differences into account, the personal energy at work framework could serve as guidance for appropriate decisions to make sure personal energy at work is obtained.

### 7.2. Limitations and Further Research

It is hoped that the theoretical framework of personal energy at work will provide guidance for scholars on future research to improve personal energy. More empirical research should be done to justify the proposed relations between antecedents, moderators, mediators and dimensions of personal energy at work.

In identifying measurements for personal energy at work, we chose to stay within the scope of vigor and thriving and their measurement tools. The antecedent themes derived from the quantitative studies of vigor and thriving, might therefore be a limited list of antecedents, other possible antecedents may also be relevant to study in the future. Future research could go beyond measurements related to personal energy and vitality, like empowerment, to find and test a proper tool that fully measures all dimensions of personal energy at work.

Personal core values are often mentioned in relation to personal energy at work [103,113,120–122] but have not (yet) been measured with regards to the spiritual dimension of personal energy. Moreover, core values are linked to the spiritual dimension of personal energy at work, however the items of vigor and thriving are not as well represented for the spiritual dimension as for the other dimensions. A deeper dive into the spiritual dimension could be done with measurement tools regarding meaningful work [72] or calling [74] that relate to core values. Therefore, further review research could concern the investigation of core values and its possible role in the theoretical framework of personal energy at work.

Future research could address spillover effects from other life domains, as the experience of energy at work is also influenced by work-life balance [60]. Scholars could learn from a more comprehensive and deeper analysis of the influence of other life domains. In addition, spillover can be found in relationships. As the data on personal factors that relate to personal energy at work, were found to be focused on the individual. Therefore, a suggestion for area of future research concerns how energy occurs at different levels of analysis, for example research on how groups can share collective energy [13,37].

In the reviewed articles the concept of identity has hardly been mentioned [123,124], while scholars have researched this and made connections to feeling energized [125]. Therefore, researching identity could provide additional knowledge on the personal factors that enhance personal energy at work.

Another topic would be to investigate the interrelation between the four dimensions of personal energy at work. Is there one dimension with significantly more impact on overall personal energy at work, or do the dimensions work synergistically? Interesting would be to research the distinct role of each energy dimension on outcomes such as health and productiveness.

We summarized our suggested directions for future research in Table 4.

**Table 4.** Summary of future research directions.

| Research Gap | Topic to Be Researched | Research Question |
|---|---|---|
| Empirical research on the construct of personal energy at work with ist antecedents and boundary conditions is lacking in literature | Construct of personal energy at work wiistits antecedents and boundary conditions | How are personal factors, external factors, strain and recovery related to the dimensions of personal energy at work? |
| A measurement tool that addresses physical, emotional, mental and spiritual energy is missing in literature | A measurement of the four dimensions of personal energy at work | How can physical, emotional, mental and spiritual energy be measured? |
| Limited knowledge of spiritual energy in relation to personal energy at work | Spiritual energy | How do meaningful work, calling and core values relate to spiritual energy? |
| Research on personal energy at work versus personal energy at home | Spillover effects life domains | What are the spillover effects of personal energy at work on personal life? |
| Limited knowledge of collective energy | Collective energy | What is the relationship between collective energy and personal energy at work? |
| Limited knowledge of working identity and feeling energized | Working identity | What is the relationship between working identity and personal energy at work? |
| Empirical research regarding the interrelationship of the four dimensions of personal energy at work is lacking in literature | Construct dimensionality of personal energy at work | What is the distinct role of each energy dimension in relation to outcomes, such as health and productiveness? |

### 7.3. Practical Implications

The theoretical framework of personal energy at work could be applied by professionals when developing people management practices that stimulate personal energy at work. This framework can support HR management in achieving sustainable employability and performance, by maintaining and stimulating personal resources of employees, even during critical times when work pressure tends to get high. Whereas traditional approaches to improve a vital workforce focused on interventions for employees to cope with workload and prevent fatigue [126], our research suggests that employees and employers work together in embedding proper behaviors like making sure everyone takes sufficient recovery brakes.

Drawing on our research, we recommend two approaches to enhance personal energy at work: individual strategies and organizational support. We see a dual responsibility to enhance personal energy at work; there is a role for the employees as well as for the employer.

The role of employees is to do a deep dive in their personality, needs, behavior and competencies. For example, they can ask for guidance to increase their resilience or explore where their core competencies lie to align them with their work. Characteristics of people, like personalities, resources, beliefs, values, cognitions and behaviors have been found to be among the strongest determinants for their psychological and physical health when faced with stressful experiences [113].

The role of the employer in their turn, is to investigate the contextual drivers and barriers of personal energy at work and develop a plan of improvement. There is a need to develop strategies for employees that are appropriate for responding to today's rapidly changing work environment and all the stressors that come with it. A plan of improvement can contain corporate environment changes and interventions to help employees enhance their four dimensions of personal energy at work. A next step for a company might be to look at opportunities of how to stimulate employees to participate in these interventions and measure the outcomes.

To improve performance, HR practices can play an important role in stimulating employees to work on their personal energy, thus there is a need to understand how HR can support [1]. Line managers and HR should be aware of what factors influence employees feeling energized and take proper actions. In order to minimize threat of occupational

stress and fatigue, HR practices should apply to the multidimensionality of personal resources and support to sustain and enhance personal energy levels [21]. Companies need to incorporate opportunities for learning and change management into job design and work scheduling, where line managers can assess employee energy levels and the degree to which they are learning at work [81]. According to Pluta and Rudawska [21], companies should focus on the level of personal resources that impact all four potentials (physical, mental, emotional and spiritual) and perform regular audits to measure this. The overall objective of a company should be to create the optimal context to stimulate employees to be aware of their personal energy dimensions and enhance them.

**Author Contributions:** Data curation, A.F.J.K.; Formal analysis, A.F.J.K.; Funding acquisition, A.F.J.K.; Investigation, A.F.J.K.; Methodology, A.F.J.K.; Project administration, A.F.J.K.; Resources, A.F.J.K.; Software, A.F.J.K.; Supervision, M.T., E.I.L. and S.N.K.; Validation, A.F.J.K., M.T., E.I.L. and S.N.K.; Visualization, A.F.J.K.; Writing—original draft, A.F.J.K. All authors have read and agreed to the published version of the manuscript.

**Funding:** This research received no external funding.

**Institutional Review Board Statement:** Not applicable.

**Informed Consent Statement:** Not applicable.

**Data Availability Statement:** Data available upon request.

**Conflicts of Interest:** The authors declare no conflict of interest.

## Appendix A

**Table A1.** Constructs and dimensions of records included in review 1.

| Author | Year | Source | Terminology in Paper | Construct | Dimensions |
|---|---|---|---|---|---|
| Abid | 2020 | [127] | Thriving | Thriving | Vitality and learning |
| Adil | 2016 | [80] | Vigor | Vigor | Physical strength, emotional energy and cognitive liveliness |
| Alcoba | 2017 | [128] | Human energy | Energy dimensions | Mental and spiritual energy |
| Asirvatham | 2019 | [129] | Vitality | Vitality | - |
| Atwater & Carmeli | 2009 | [53] | Feelings of energy | Energy | - |
| Baker | 2019 | [25] | Emotional energy | Energy dimensions | Emotional energy |
| Baker | 2018 | [130] | Physical and mental capacity | Energy dimensions | Physical and mental energy |
| Bakker | 2008 | [131] | Work engagement | Vigor as part of engagement | - |
| Bakker | 2011 | [58] | Work engagement | Vigor as part of engagement | - |

**Table A1.** *Cont.*

| Author | Year | Source | Terminology in Paper | Construct | Dimensions |
|---|---|---|---|---|---|
| Bakker | 2020 | [132] | Proactive vitality management | Vitality | |
| Barber | 2013 | [133] | Work engagement | Vigor as part of engagement | - |
| Barber | 2017 | [134] | Sleep quality, exercise | Energy dimensions | Physical energy |
| Basinska | 2020 | [32] | Thriving | Thriving | Vitality and learning |
| Bensemmane | 2018 | [135] | Thriving | Thriving | Vitality and learning |
| Bertrams | 2021 | [136] | Vitality | Vitality | - |
| Bickerton | 2014 | [50] | Work engagement | Vigor as part of engagement | - |
| Blanco-Donoso | 2019 | [137] | Vigor | Vigor as part of engagement | - |
| Brosch | 2018 | [138] | Vigor | Vigor | Physical strength, emotional energy and cognitive liveliness |
| Cangiano | 2019 | [139] | Vitality | Vigor as part of engagement | - |
| Carmeli | 2016 | [33] | Thriving | Thriving | Vitality and learning |
| Carmeli & Spreitzer | 2009 | [35] | Thriving | Thriving | Vitality and learning |
| Carvalho | 2018 | [140] | Mental health | Energy dimensions | Mental energy |
| Celestine | 2020 | [141] | Energy management | Energy | - |
| Cham | 2021 | [142] | Energy Management | Energy dimensions | Biological processes (like sleep), psychophysiological states (like arousal), self-regulatory processes (like goal setting) |
| Chen | 2016 | [143] | Personal engagement | Vigor as part of engagement | - |
| Christian | 2015 | [144] | Work engagement | Vigor as part of engagement | - |
| Chuang | 2021 | [145] | Vitality | Subjective vitality | - |

**Table A1.** *Cont.*

| Author | Year | Source | Terminology in Paper | Construct | Dimensions |
|---|---|---|---|---|---|
| Cole | 2012 | [18] | Productive energy | Productive energy | Affective, cognitive and behavioral |
| Conde | 2021 | [146] | Energy levels | Energy | - |
| Costa | 2016 | [147] | Work engagement | Vigor as part of engagement | - |
| Costa | 2014 | [51] | Work engagement | Vigor as part of engagement | - |
| Covey | 1989 | [79] | Energy | Energy dimensions | Physical, emotional, mental and spiritual energy |
| Crain | 2018 | [148] | Human energy | Energy dimensions | Physical, emotional, mental and spiritual energy |
| Cullen | 2018 | [61] | Thriving | Thriving | Vitality and learning |
| de Bloom | 2010 | [54] | Energy | Energy | - |
| de Clercq | 2019 | [149] | Positive job energy | Energy dimensions | Emotional energy |
| de Jonge | 2019 | [2] | Vitality | Vitality | - |
| Deci | 2008 | [28] | Vitality | Vitality | - |
| Delegach | 2021 | [150] | Vitality | Vitality | - |
| Drain | 2016 | [151] | Physical demands | Energy dimensions | Physical energy |
| Dubreuil | 2014 | [63] | Subjective vitality | Subjective vitality | - |
| Eberly | 2017 | [152] | Availability of time/energy resources | Energy | - |
| Fein | 2015 | [153] | Energy exchange between work and non-work | Energy | - |
| Franco-Santos | 2017 | [154] | Thriving | Thriving | Vitality and learning |
| Fritz | 2011 | [23] | Human energy at work | Energy dimensions | Physical, emotional, mental and spiritual energy |
| Gabriel | 2020 | [155] | Psychological vitality | Energy dimensions | Emotional energy |
| Gerbasi | 2015 | [156] | Thriving | Thriving | Vitality and learning |
| Grawitch | 2015 | [15] | Energy resources | Energy | - |

**Table A1.** *Cont.*

| Author | Year | Source | Terminology in Paper | Construct | Dimensions |
|---|---|---|---|---|---|
| Gucciardi | 2015 | [157] | Thriving | Thriving | Vitality and learning |
| Hahn | 2012 | [64] | Subjective vitality | Subjective vitality | - |
| Haynie | 2017 | [158] | Job engagement | Vigor as part of engagement | - |
| Hobfoll | 1989 | [103] | Energy resources | Energy | - |
| Huber | 2017 | [65] | Subjective vitality | Subjective vitality | - |
| Jabeen | 2021 | [159] | Vitality | Vitality | - |
| Janicke-Bowles | 2019 | [160] | Vitality | Vitality | - |
| Johnston | 2019 | [161] | Physical Energy Expenditure | Energy dimensions | Physical energy |
| Joki | 2017 | [162] | Weight management | Energy dimensions | Physical energy |
| Kahrobaei | 2016 | [36] | Collective energy | Productive energy | Affective, cognitive and behavioral |
| Kanning | 2013 | [163] | Energetic arousal | Energy | - |
| Kark | 2009 | [164] | Vitality | Vitality | - |
| Karkkola | 2019 | [165] | vitality | Vigor as part of engagement | - |
| Kipfelsberger | 2019 | [37] | Productive energy | Productive energy | Affective, cognitive and behavioral |
| Kira | 2014 | [122] | Thriving | Thriving | Vitality and learning |
| Kleine | 2019 | [24] | Thriving | Thriving | Vitality and learning |
| Klotz | 2021 | [166] | Energy reserves | Energy dimensions | Cognitive, emotional, prosocial, and physical energy |
| Kruglanski | 2012 | [167] | Energy | Energy dimensions | Mental energy |
| Kujanpaa | 2021 | [168] | Vitality | Subjective vitality | - |
| Lam | 2014 | [169] | Thriving | Thriving | Vitality and learning |
| Lam | 2016 | [30] | Vigor | Vigor | Physical strength, emotional energy and cognitive liveliness |
| Lanaj | 2019 | [170] | Cognitive energetics theory | Energy dimensions | Mental energy |

**Table A1.** *Cont.*

| Author | Year | Source | Terminology in Paper | Construct | Dimensions |
|---|---|---|---|---|---|
| Lee | 2016 | [171] | Physical energy | Energy dimensions | Physical energy |
| Lemmon | 2020 | [172] | Engagement | Vigor as part of engagement | - |
| Liebhart | 2013 | [173] | Productive energy | Energy dimensions | Physical and mental energy |
| Liu | 2021 | [174] | Vitality | Vitality | - |
| Loch | 2020 | [175] | Energy | Energy dimensions | Mental fatigue, mental recovery |
| Loehr & Schwartz | 2003 | [56] | Energy | Energy dimensions | Physical, emotional, mental and spiritual energy |
| Lukyanchenko | 2016 | [176] | Psychological wellbeing | Energy dimensions | Emotional energy |
| Lupano | 2018 | [52] | Work engagement | Vigor as part of engagement | - |
| Marques | 2021 | [45] | Energy | Energy dimensions | physical, cognitive, social/spiritual, and psycho-emotional |
| McCarthy | 2014 | [177] | Food and physical activity patterns | Energy dimensions | Physical energy |
| Mitchell | 2019 | [178] | Engagement | Vigor as part of engagement | - |
| Mosteo | 2016 | [179] | Goal directed energy | Energy dimensions | Emotional energy |
| Mushtaq | 2017 | [180] | Thriving | Thriving | Vitality and learning |
| Nawaz | 2020 | [181] | Thriving | Thriving | Vitality and learning |
| Newton | 2020 | [19] | Energy | Energy dimensions | Physical, cognitive and emotional energy |
| Nie | 2021 | [182] | Energy management | Energy | - |
| Niessen | 2017 | [26] | Thriving | Thriving | Vitality and learning |
| Niessen | 2012 | [3] | Thriving | Thriving | Vitality and learning |
| Nixon | 1982 | [183] | Energy | Energy dimensions | Physical and mental energy |
| Oliveira | 2021 | [184] | Thriving | Thriving | Learning, vitality |

**Table A1.** *Cont.*

| Author | Year | Source | Terminology in Paper | Construct | Dimensions |
|---|---|---|---|---|---|
| Op den Kamp | 2018 | [62] | Proactive vitality management | Vitality | - |
| Owens | 2016 | [185] | Relational energy | Energy dimensions | Emotional energy |
| Ozyilmaz | 2019 | [59] | Engagement | Vigor as part of engagement | - |
| Paterson | 2014 | [81] | Thriving | Thriving | Vitality and learning |
| Pitesa | 2018 | [186] | Physical demands at work | Energy dimensions | Physical energy |
| Pluta | 2016 | [21] | Energies available by an individual | Energy dimensions | Physical, emotional, mental and spiritual energy |
| Pommier | 2018 | [187] | Physical or psychic energy | Energy dimensions | Physical, emotional and mental energy |
| Porath | 2012 | [12] | Thriving | Thriving | Vitality and learning |
| Quinn | 2012 | [6] | Human energy | Energy dimensions | Physical, emotional, mental and spiritual energy |
| Quinn & Dutton | 2005 | [76] | Energy | Energy | - |
| Rattrie | 2020 | [188] | Energy | Energy | - |
| Rego | 2019 | [189] | Team positive energizing | Energy | - |
| Russo | 2016 | [55] | Employee energy | Energy | - |
| Ryan & Frederick | 1997 | [8] | Subjective vitality | Subjective Vitality | - |
| Sanz-Vergel | 2010 | [60] | Vigor | Vigor as part of engagement | - |
| Schippers | 2011 | [20] | Energy | Energy dimensions | Physical, emotional and mental energy |
| Schwartz | 2007 | [22] | Energy | Energy dimensions | Physical, emotional, mental and spiritual energy |

**Table A1.** *Cont.*

| Author | Year | Source | Terminology in Paper | Construct | Dimensions |
|---|---|---|---|---|---|
| Shirom | 2011 | [104] | Vigor | Vigor | Physical strength, emotional energy and cognitive liveliness |
| Shirom | 2004 | [9] | Vigor | Vigor | Physical strength, emotional energy and cognitive liveliness |
| Skare | 2016 | [190] | Vitality | Vitality | - |
| Solat | 2020 | [191] | Energy resources | Energy | - |
| Sonnentag | 2008 | [49] | Vigor | Vigor | Physical strength, emotional energy and cognitive liveliness |
| Spanouli | 2020 | [192] | Vitality | Vitality | - |
| Spreitzer | 2005 | [13] | Thriving | Thriving | Vitality and learning |
| Standal | 2021 | [193] | Energy | Energy | - |
| Stephan | 2020 | [194] | Subjective vitality | Subjective vitality | - |
| Ten Brummelhuis | 2016 | [195] | Engagement | Vigor as part of engagement | - |
| Ten Brummelhuis | 2012 | [196] | Personal resources | Energy dimensions | Physical, emotional and mental energy |
| Tummers | 2015 | [116] | Vitality | Vitality | - |
| Tummers | 2018 | [1] | Vitality | Vitality | - |
| van der Walt | 2018 | [34] | Thriving | Thriving | Vitality and learning |
| van Hooff | 2019 | [197] | Positive energy | Energy dimensions | Emotional energy |
| van Hooff | 2017 | [198] | Vigor | Vigor | Physical strength, emotional energy and cognitive liveliness |
| van Ruysseveldt | 2011 | [199] | Job resources | Energy dimensions | Emotional and mental energy |
| van Wyk | 2016 | [200] | Positive emotions | Energy dimensions | Emotional energy |
| van Zyl | 2021 | [201] | Work engagement | Vigor as part of engagement | |
| Venz | 2015 | [202] | Work engagement | Vigor as part of engagement | - |

**Table A1.** *Cont.*

| Author | Year | Source | Terminology in Paper | Construct | Dimensions |
|---|---|---|---|---|---|
| Wallace | 2016 | [90] | Thriving | Thriving | Vitality and learning |
| Weigelt | 2021 | [203] | Human energy | Energy | - |
| Welbourne | 2014 | [204] | Employee energy | Energy dimensions | Physical energy |
| Wörtler | 2020 | [205] | Vitality | Vitality | - |
| Wray-Lake | 2019 | [206] | Subjective vitality | Subjective vitality | - |
| Xiong | 2020 | [207] | Work engagement | Vigor as part of engagement | |
| Zhang | 2021 | [208] | Vitality | Vitality | - |
| Zhou | 2020 | [209] | Energy | Energy | - |

## Appendix B

**Table A2.** Items from vigor and thriving measurements allocated to personal energy at work dimensions.

| Construct | Item | Personal Energy Dimension | Rationale |
|---|---|---|---|
| Vigor | I feel full of pep | Physical energy | As defined by Shirom [9] to be related to physical strength, representing the affective feeling in the body to feel capable to do work |
| Vigor | I feel I have physical strength | Physical energy | As defined by Shirom [9] to be related to physical strength, representing the kinetic energy in the body |
| Vigor | I feel vigorous | Physical energy | As defined by Shirom [9] to be related to physical strength, representing the affective feeling in the body to feel capable to do work |
| Vigor | I feel energetic | Physical energy | As defined by Shirom [9] to be related to physical strength, representing the affective feeling in the body to feel capable to do work |
| Vigor | A feeling of vitality | Physical energy | As defined by Shirom [9] to be related to physical strength, representing the affective feeling in the body to feel capable to do work |
| Vigor | I feel mentally alert | Mental energy | As defined by Shirom [9] to be related to cognitive liveliness, representing the mental capacity not to be distracted easily |
| Vigor | I feel I can think rapidly | Mental energy | As defined by Shirom [9] to be related to cognitive liveliness, representing the focus to make conscious decisions |
| Vigor | I feel I am able to contribute new ideas | Mental energy | As defined by Shirom [9] to be related to cognitive liveliness, representing the mental capacity of showing creative skills |

**Table A2.** *Cont.*

| Construct | Item | Personal Energy Dimension | Rationale |
|---|---|---|---|
| Vigor | I feel able to be creative | Mental energy | As defined by Shirom [9] to be related to cognitive liveliness, representing the mental capacity of showing creative skills |
| Vigor | A feeling of flow | Mental energy | As defined by Shirom [9] to be related to cognitive liveliness, representing the focus to stick to an assignment as needed |
| Vigor | I feel able to show warmth to others | Emotional energy | As defined by Shirom [9] to be related to emotional energy, representing empathy towards colleagues |
| Vigor | I feel able to be sensitive to the needs of coworkers and customers | Emotional energy | As defined by Shirom [9] to be related to emotional energy, representing empathy towards colleagues |
| Vigor | I feel I am capable of investing emotionally in coworkers and customers | Emotional energy | As defined by Shirom [9] to be related to emotional energy, representing the positive mindset to be able to connect with colleagues |
| Vigor | I feel capable of being sympathetic to coworkers and customers | Emotional energy | As defined by Shirom [9] to be related to emotional energy, representing the positive mindset to support colleagues |
| Thriving | I find myself learning often | Mental energy | As defined by Porath [12] to be related to the learning dimension, representing the cognitive development |
| Thriving | I continue to learn more as time goes by | Mental energy | As defined by Porath [12] to be related to the learning dimension, representing the cognitive development |
| Thriving | I see myself continually improving | Mental energy | As defined by Porath [12] to be related to the learning dimension, representing the cognitive development |
| Thriving | I am not learning | Mental energy | As defined by Porath [12] to be related to the learning dimension, representing the lack of gaining skills |
| Thriving | I am developing a lot as a person | Mental energy | As defined by Porath [12] to be related to the learning dimension, representing the cognitive development |
| Thriving | I feel alive and vital | Physical energy | As defined by Porath [12] to be related to the vitality dimension, representing the physical affective feeling to do the work |
| Thriving | I have energy and spirit | Spiritual energy | As defined by Porath [12] to be related to the vitality dimension, representing spiritual state of believing in your own capacity |
| Thriving | I do not feel very energetic | Physical energy | As defined by Porath [12] to be related to the vitality dimension, representing the lack physical kinetic energy that is needed to do the work |

**Table A2.** *Cont.*

| Construct | Item | Personal Energy Dimension | Rationale |
| --- | --- | --- | --- |
| Thriving | I feel alert and awake | Physical energy | As defined by Porath [12] to be related to the vitality dimension, representing to have rested enough and recharged to do the work |
| Thriving | I am looking forward to each new day | Spiritual energy | As defined by Porath [12] to be related to the vitality dimension, representing spirit of finding purpose in every day |

## Appendix C

**Table A3.** Antecedent themes of vigor and thriving of records included in review 2.

| | Year | Source | Independent Variable (IV) | | | | | Relationship between IV and DV Environment | Moderator | Dependent Variable (DV) |
| --- | --- | --- | --- | --- | --- | --- | --- | --- | --- | --- |
| | | | Competence | Behavior | Psychological State | Personality | Supervisor | | | |
| Cheng | 2014 | [210] | Coping strategies | | | | | + | | Vigor |
| Casper | 2017 | [211] | Approach-coping efforts | | | | | + | | Vigor |
| Hoppe | 2017 | [212] | Job control | | | | | + | | Vigor |
| Wefald | 2017 | [213] | Performance | | | | | + | | Vigor |
| Wefald | 2017 | [213] | Customer service orientation | | | | | + | | Vigor |
| Armon | 2011 | [86] | | Physical exercise (control variable) | | | | + | | Vigor |
| Armon | 2011 | [86] | | Self rated health (control variable) | | | | + | | Vigor |
| Niessen | 2012 | [3] | | Task Focus | | | | + | | Vigor |
| Niessen | 2012 | [3] | | Heedful relating | | | | + | | Vigor |
| Niessen | 2012 | [3] | | Exploration | | | | + | | Vigor |
| Kung | 2014 | [214] | | Positive perfectionism | | | | + | | Vigor |
| Kung | 2014 | [214] | | Negative perfectionism | | | | - | | Vigor |
| Oerlemans | 2014 | [93] | | Off-job time spent on social activities | | | | + | Burnout (-) | Vigor |
| Oerlemans | 2014 | [93] | | Off-job time spent on physical activities | | | | + | | Vigor |
| Oerlemans | 2014 | [93] | | Off-job time spent on low effort activities | | | | + | Burnout (-) | Vigor |
| Oerlemans | 2014 | [93] | | Off job time spent on work-related activities, for people high in burn out | | | | - | | Vigor |
| Oerlemans | 2014 | [93] | | Off job time spent on low effort (social, physical) activities, for people high in burn out | | | | + | | Vigor |
| Op den Kamp | 2018 | [62] | | Proactive vitality management (PVM) | | | | + | | Vigor |
| Bakker | 2019 | [215] | | Surface acting at home | | | | - | | Vigor |
| Pulido-Martos | 2020 | [216] | | Physical activity | | | | + | | Vigor |
| Ginoux | 2021 | [217] | | Recovery experience activities | | | | + | | Vigor |
| Carmeli | 2009 | [218] | | | Bonding social capital | | | + | | Vigor |
| Little | 2011 | [83] | | | Secure attachment | | | + | | Vigor |
| Little | 2011 | [83] | | | Counterdependence | | | - | | Vigor |
| Little | 2011 | [83] | | | Overdependence | | | - | | Vigor |
| Armon | 2011 | [86] | | | Emotional stability | | | + | | Vigor |
| Sanz-Vergel | 2011 | [219] | | | Daily detachment from work for people with high level of home role salience | | | + | | Vigor |
| Sanz-Vergel | 2011 | [219] | | | Daily detachment from work for people with low level of home role salience | | | - | | Vigor |
| Oerlemans | 2014 | [93] | | | Burn out | | | - | | Vigor |
| Wefald | 2017 | [213] | | | Job satisfaction | | | + | | Vigor |
| Wefald | 2017 | [213] | | | Engagement | | | + | | Vigor |
| Wefald | 2017 | [213] | | | Job involvement | | | + | | Vigor |
| di Luzio | 2019 | [220] | | | Satisfaction of need for relatedness | | | + | | Vigor |
| Casper | 2019 | [221] | | | Positive work reflections | | | + | | Vigor |
| Casper | 2019 | [221] | | | Negative work reflections | | | - | | Vigor |
| David | 2020 | [222] | | | Self-verification striving | | | + | | Vigor |
| Armon | 2011 | [86] | | | | Conscientiousness | | + | | Vigor |
| Armon | 2011 | [86] | | | | Agreeableness | | + | | Vigor |
| Armon | 2011 | [86] | | | | Extraversion | | + | | Vigor |
| Armon | 2011 | [86] | | | | Openness | | + | | Vigor |
| Adil | 2016 | [29] | | | | | Leader-Member exchange | + | | Vigor |
| Hoppe | 2017 | [212] | | | | | Supervisor support | + | | Vigor |
| Swords | 2017 | [223] | | | | | Supervisory working alliance | + | | Vigor |

**Table A3.** *Cont.*

| | Year | Source | Independent Variable (IV) | | | | | | Relationship between IV and DV | Moderator | Dependent Variable (DV) |
|---|---|---|---|---|---|---|---|---|---|---|---|
| | | | Competence | Behavior | Psychological State | Personality | Supervisor | Environment | | | |
| Parent-Rocheleau | 2020 | [224] | | | | | Leaders' energy | | + | Leader-member exchange (+) | Vigor |
| Tremblay | 2021 | [225] | | | | | Leader-member exchange | | + | | Vigor |
| Cheng | 2014 | [210] | | | | | | Job insecurity | - | | Vigor |
| Aikens | 2014 | [89] | | | | | | Mindfulness intervention | + | | Vigor |
| Ben-Hador | 2016 | [226] | | | | | | Intra-organizational social capital | + | | Vigor |
| Mauno | 2017 | [227] | | | | | | Work-home conflict | - | Co-worker support (-) | Vigor |
| Swords | 2017 | [223] | | | | | | Collective set of pressure, threat, financial strain, relationship conflict and supervisory working | - | | Vigor |
| Swords | 2017 | [223] | | | | | | General work-related threats | - | | Vigor |
| Wefald | 2017 | [213] | | | | | | Perceived organizational support | + | | Vigor |
| Bernez | 2018 | [228] | | | | | | Presence of a garden | + | | Vigor |
| Dugan | 2020 | [229] | | | | | | Second shift workload | - | | Vigor |
| Cai | 2020 | [230] | | | | | | Social capital | + | | Vigor |
| van Iperen | 2020 | [94] | | | | | | Running-related demands | - | Running related recovery (-), Running-related resources (-) | Vigor |
| Pulido-Martos | 2021 | [231] | | | | | | Social support | + | Telework decrease (+) | Vigor |
| Balk | 2021 | [232] | | | | | | Daily sport demands | - | | Vigor |
| de Jonge | 2021 | [233] | | | | | | Cognitive resources | + | Cognitive resources (+) | Vigor |
| de Jonge | 2021 | [233] | | | | | | Cognitive demands | + | | Vigor |
| de Jonge | 2021 | [233] | | | | | | Emotional demands | - | | Vigor |
| de Jonge | 2021 | [233] | | | | | | Emotional resources | + | | Vigor |
| de Jonge | 2021 | [233] | | | | | | Physical demands | + | | Vigor |
| Duan | 2021 | [234] | | | | | | Employment relationship (permanent vs. temporary) | + | Need for status (+) | Vigor |
| Venz | 2021 | [235] | | | | | | Perceived morning weather | - | Weather sensitivity (+) | Vigor |
| Gucciardi | 2015 | [236] | Mental toughness | | | | | | + | | Thriving |
| Gucciardi | 2015 | [236] | Academic goal progress | | | | | | + | | Thriving |
| Gucciardi | 2015 | [236] | Social goal progress | | | | | | + | | Thriving |
| Gucciardi | 2017 | [237] | Mental toughness | | | | | | + | | Thriving |
| Bensemmane | 2018 | [135] | Self-efficacy | | | | | | + | | Thriving |
| Taneva | 2018 | [238] | Optimization | | | | | | + | | Thriving |
| Cullen | 2018 | [61] | Political skill | | | | | | + | | Thriving |
| Zhu, 2019 | 2019 | [239] | Job self-efficacy | | | | | | + | | Thriving |
| Weigelt | 2019 | [240] | Problem-solving pondering | | | | | | + | | Thriving |
| Christensen-Salem | 2020 | [241] | Creative self-efficacy | | | | | | + | | Thriving |
| Mahomed | 2020 | [242] | Strength use | | | | | | + | | Thriving |
| Ding | 2020 | [243] | Self-efficacy | | | | | | + | | Thriving |
| Ding | 2020 | [243] | Strengths use | | | | | | + | | Thriving |
| Mansur | 2020 | [244] | Career adaptability | | | | | | + | | Thriving |
| Zhang | 2020 | [245] | Social functioning | | | | | | + | | Thriving |
| Abid | 2020 | [127] | Self-efficacy | | | | | | + | | Thriving |
| Han | 2021 | [246] | Overqualification | | | | | | - | | Thriving |
| Moore | 2021 | [247] | Strengths use at work | | | | | | + | | Thriving |
| Zhou | 2021 | [248] | Academic personal best goal (goals that match the academic context) | | | | | | + | | Thriving |
| Wang | 2021 | [249] | Feedback-seeking from team members | | | | | | + | Mindfulness (+) | Thriving |
| Ren | 2021 | [250] | Cultural intelligence | | | | | | + | | Thriving |
| Yang | 2021 | [251] | Self-efficacy | | | | | | + | | Thriving |
| Liu | 2022 | [252] | General self-efficacy | | | | | | + | | Thriving |
| Paterson | 2014 | [81] | | Heedful relating | | | | | + | | Thriving |

**Table A3.** *Cont.*

| | Year | Source | Independent Variable (IV) | | | | | Relationship between IV and DV Environment | Moderator | Dependent Variable (DV) |
|---|---|---|---|---|---|---|---|---|---|---|
| | | | Competence | Behavior | Psychological State | Personality | Supervisor | | | |
| Paterson | 2014 | [81] | | Task focus | | | | + | | Thriving |
| Wallace | 2016 | [90] | | Promotion focus | | | | + | Employee involvement climate (+) | Thriving |
| Wallace | 2016 | [90] | | Prevention focus | | | | - | | Thriving |
| Abid | 2016 | [80] | | Heedful relating | | | | + | | Thriving |
| Wu | 2019 | [109] | | Collective mindfulness | | | | + | | Thriving |
| Jiang | 2019 | [253] | | Learning goal orientation | | | | + | | Thriving |
| Jiang | 2019 | [253] | | Exploration at work | | | | + | | Thriving |
| Paleari | 2019 | [254] | | Counterproductive work behaviors | | | | - | | Thriving |
| Aryee | 2019 | [255] | | Customer orientation | | | | + | | Thriving |
| Rego | 2021 | [256] | | Employees' self-attributed grit (goals) | | | | + | Perceived leader support (+) | Thriving |
| Riaz | 2020 | [257] | | Heedful relating | | | | + | | Thriving |
| Chen | 2020 | [258] | | Online leisure crafting | | | | + | | Thriving |
| Chen | 2020 | [259] | | Voice behavior (speaking out) | | | | + | | Thriving |
| Zhang | 2020 | [260] | | Employee helping behavior | | | | + | | Thriving |
| Xu | 2020 | [261] | | Taking charge | | | | + | Leader's role ambiguity (+) | Thriving |
| Sahin | 2020 | [108] | | Mindfulness practice | | | | + | | Thriving |
| Guan | 2020 | [78] | | Job crafting | | | | + | | Thriving |
| Fritz | 2021 | [262] | | Anticipated task focus in the morning | | | | + | | Thriving |
| Usman | 2021 | [263] | | Heedful relating | | | | + | | Thriving |
| Usman | 2021 | [263] | | Task focus | | | | + | | Thriving |
| Usman | 2021 | [264] | | Task focus | | | | + | | Thriving |
| Usman | 2021 | [264] | | Heedful relating | | | | + | | Thriving |
| Han | 2021 | [246] | | Job crafting toward strengths and interests | | | | + | Overqualification (+) | Thriving |
| Gucciardi | 2015 | [157] | | | Believing in the incremental theory of mental toughness (vs ambivalent theory) | | | + | | Thriving |
| Kabat-Farr | 2017 | [265] | | | Self assurance in response to Interpersonal Citizenship Behavior | | | + | | Thriving |
| Mushtaq | 2017 | [180] | | | Fairness perception | | | + | | Thriving |
| Zhao | 2018 | [266] | | | Job satisfaction | | | + | | Thriving |
| Abid | 2018 | [267] | | | Prosocial motivation | | | + | | Thriving |
| Ehrnhardt | 2019 | [84] | | | Relational attachment | | | + | | Thriving |
| Weigelt | 2019 | [240] | | | Psychological detachment | | | + | | Thriving |
| Weigelt | 2019 | [240] | | | Affective rumination | | | - | | Thriving |
| Weigelt | 2019 | [240] | | | Positive work reflection | | | + | | Thriving |
| Weigelt | 2019 | [240] | | | Negative work reflection | | | - | | Thriving |
| Liu | 2019 | [268] | | | Employees' Paradox Mindset | | | + | Leaders' Paradox Mindset (+) | Thriving |
| Yang | 2019 | [269] | | | Future work self | | | + | Overall fairness of supervisor (+) | Thriving |
| Kim | 2020 | [270] | | | Psychological empowerment | | | + | | Thriving |
| Kim | 2020 | [270] | | | Organization based self esteem | | | + | | Thriving |
| Mahomed | 2020 | [242] | | | Autonomy satisfaction | | | + | | Thriving |
| Abid | 2020 | [127] | | | Prosocial motivation | | | + | | Thriving |
| Abid | 2020 | [271] | | | Fairness perception | | | + | | Thriving |
| Guan | 2020 | [272] | | | Work meaningfulness | | | + | | Thriving |
| Nawaz | 2020 | [181] | | | Prosocial motivation | | | + | | Thriving |
| Nawaz | 2020 | [181] | | | Psychological capital | | | + | | Thriving |
| Guan | 2020 | [78] | | | Meaningfulness | | | + | | Thriving |
| Zhang | 2020 | [260] | | | Psychological availability | | | + | | Thriving |
| Cain | 2020 | [273] | | | Having a calling | | | + | | Thriving |
| Lin | 2020 | [106] | | | Perceived challenge stressors | | | + | Supervisor developmental feedback (+) | Thriving |
| Lin | 2020 | [106] | | | Perceived hindrance stressors | | | - | | Thriving |
| Sahin | 2021 | [108] | | | Psychological well-being | | | + | | Thriving |
| Fritz | 2021 | [262] | | | Reattachment to work | | | + | | Thriving |
| Fritz | 2021 | [262] | | | Activated positive affect | | | + | | Thriving |
| Guo | 2021 | [274] | | | Intrinsic motivation | | | + | | Thriving |

**Table A3.** *Cont.*

| | Year | Source | Independent Variable (IV) | | | | | Relationship between IV and DV Environment | Moderator | Dependent Variable (DV) |
|---|---|---|---|---|---|---|---|---|---|---|
| | | | Competence | Behavior | Psychological State | Personality | Supervisor | | | |
| Pace | 2021 | [275] | | | Perceived employability | | | + | | Thriving |
| Zhou | 2021 | [248] | | | Harmonious passion | | | + | | Thriving |
| Zhou | 2021 | [248] | | | Obsessive passion | | | + | | Thriving |
| Ozcan | 2021 | [276] | | | Mindfulness | | | + | | Thriving |
| Oliveira | 2021 | [184] | | | Focus on opportunities | | | + | | Thriving |
| Oliveira | 2021 | [184] | | | Occupational future time perspective | | | + | | Thriving |
| Zhu | 2021 | [277] | | | Psychological capital | | | + | | Thriving |
| Rahaman | 2021 | [278] | | | Interpersonal justice (feeling fair about company methods) | | | + | | Thriving |
| Zhang | 2021 | [279] | | | Perceived insider status (feeling of belonging) | | | + | Proactive personality (+) | Thriving |
| Kinoshita | 2021 | [280] | | | Eudaimonic motives | | | + | | Thriving |
| Kinoshita | 2021 | [280] | | | Hedonic motives | | | + | | Thriving |
| Kinoshita | 2021 | [280] | | | Basic Psychological Need Satisfaction | | | + | | Thriving |
| Yang | 2021 | [251] | | | Positive affect | | | + | | Thriving |
| Liu | 2022 | [252] | | | Performance pressure | | | + | | Thriving |
| Hennekam | 2017 | [85] | | | | Neurotism | | - | | Thriving |
| Hennekam | 2017 | [85] | | | | Extraversion | | + | | Thriving |
| Hennekam | 2017 | [85] | | | | Conscientiousness | | + | | Thriving |
| Jiang | 2017 | [124] | | | | Proactive personality | | + | | Thriving |
| Mushtaq | 2017 | [180] | | | | Proactive personality | | + | | Thriving |
| Walumbwa | 2018 | [281] | | | | Core self-evaluation | | + | | Thriving |
| Alikaj | 2020 | [282] | | | | Proactive personality | | + | Perceived presence of high-involvement HR practices (+) | Thriving |
| Abid | 2021 | [283] | | | | Personality trait: hope | | + | | Thriving |
| Abid | 2021 | [283] | | | | Personality trait: optimism | | + | | Thriving |
| Li | 2016 | [284] | | | | | Empowering leadership | + | Employees' autonomy orientation (+) | Thriving |
| Mortier | 2016 | [67] | | | | | Managers' authentic leadership | + | | Thriving |
| Mortier | 2016 | [67] | | | | | Managers' empathy | + | | Thriving |
| Prem | 2017 | [285] | | | | | Challenge appraisal | + | | Thriving |
| Prem | 2017 | [285] | | | | | Hindrance appraisal | - | | Thriving |
| Franco-Santos | 2017 | [286] | | | | | Directive performance management | - | | Thriving |
| Franco-Santos | 2017 | [286] | | | | | Enabling performance management | + | | Thriving |
| Gucciardi | 2017 | [237] | | | | | Interpersonal controlling style of coach | - | | Thriving |
| Munc | 2017 | [287] | | | | | Provision of supervisor support | + | | Thriving |
| Niessen | 2017 | [26] | | | | | Perceived transformational leadership | + | High emotional exhaustion (-) | Thriving |
| Xu | 2017 | [88] | | | | | Leader-member exchange | + | Authentic leadership (+) | Thriving |
| Hildenbrand | 2018 | [288] | | | | | Transformational leadership | + | Openness to experience (+) | Thriving |
| Raza | 2018 | [289] | | | | | Managerial coaching | + | Perceptions of organizational politics (-) | Thriving |
| Cullen | 2018 | [61] | | | | | Role overload | - | | Thriving |
| Cullen | 2018 | [61] | | | | | Role ambiguity | - | | Thriving |
| Walumbwa | 2018 | [281] | | | | | Servant leadership | + | | Thriving |
| Ali | 2018 | [290] | | | | | Empowering leadership behavior | + | | Thriving |
| Marchiondo | 2018 | [291] | | | | | Negative appraisal | - | | Thriving |
| Marchiondo | 2018 | [291] | | | | | Positive appraisal | + | | Thriving |
| Silén | 2019 | [292] | | | | | Structural empowerment | + | | Thriving |

**Table A3.** *Cont.*

| | Year | Source | Competence | Behavior | Psychological State | Personality | Supervisor | Environment | Relationship between IV and DV | Moderator | Dependent Variable (DV) |
|---|---|---|---|---|---|---|---|---|---|---|---|
| Jiang | 2019 | [253] | | | | | Role ambiguity | | - | | Thriving |
| Li | 2019 | [293] | | | | | Leader inclusiveness | | + | Employees' regulatory focus (+) | Thriving |
| Wang | 2019 | [294] | | | | | Servant leadership | | + | Team reflexivity (+) | Thriving |
| Xu | 2019 | [295] | | | | | Leader-member exchange | | + | Store spatial crowding (-) | Thriving |
| Aryee | 2019 | [255] | | | | | Team level empowering leadership | | + | Customer orientation (-) | Thriving |
| Zeng | 2020 | [296] | | | | | Inclusive leadership | | + | | Thriving |
| Zhai | 2020 | [297] | | | | | Supervisor support | | + | | Thriving |
| Jo | 2020 | [298] | | | | | Service leadership | | + | | Thriving |
| Jiang | 2020 | [92] | | | | | Task identity | | + | Mentoring (+) | Thriving |
| Jiang | 2020 | [92] | | | | | Autonomy | | + | Mentoring (+) | Thriving |
| Raza | 2020 | [299] | | | | | Managerial coaching | | + | | Thriving |
| Chen | 2020 | [259] | | | | | Perceived leader's helping behavior | | + | | Thriving |
| Guan | 2020 | [272] | | | | | Leader-member exchange | | + | | Thriving |
| Zeng | 2020 | [300] | | | | | Mentoring | | + | Promotion focus (+) | Thriving |
| Iqbal | 2020 | [301] | | | | | Servant leadership | | + | | Thriving |
| Lin | 2020 | [106] | | | | | Transformational leadership | | + | | Thriving |
| Chang | 2020 | [302] | | | | | Authentic leadership | | + | Psychological contract (+) | Thriving |
| Khan | 2020 | [303] | | | | | Transformational leadership | | + | | Thriving |
| Usman | 2020 | [304] | | | | | Ambidextrous leadership | | + | | Thriving |
| Sahin | 2021 | [305] | | | | | Family supportive supervisor behaviors | | + | | Thriving |
| Guo | 2021 | [274] | | | | | Zhongyong leadership (integrating the opinions of all parties and pursuing harmony) | | + | | Thriving |
| Wang | 2021 | [306] | | | | | Challenge appraisal | | + | | Thriving |
| Wang | 2021 | [306] | | | | | Hindranse appraisal | | - | | Thriving |
| Usman | 2021 | [307] | | | | | Participative leadership | | + | | Thriving |
| Usman | 2021 | [263] | | | | | Servant leadership | | + | Core self-evaluations (+) | Thriving |
| Yang | 2021 | [251] | | | | | Paradoxical (fair) leader behavior | | + | | Thriving |
| Zhu | 2021 | [277] | | | | | Authentic leadership | | + | | Thriving |
| Zhang | 2021 | [279] | | | | | Differential leadership | | + | Proactive personality (+) | Thriving |
| Usman | 2021 | [264] | | | | | Abusive supervision | | - | Core self-evaluations (-) | Thriving |
| Ren | 2015 | [308] | | | | | | Job deprivation on cultural instruction competence | - | | Thriving |
| Flinchbauch | 2015 | [107] | | | | | | External hindrance stressors | - | Resilience (-) | Thriving |
| Flinchbauch | 2015 | [107] | | | | | | External challenge stressors | + | | Thriving |
| Wallace | 2016 | [90] | | | | | | Employee involvement climate | + | | Thriving |
| Abid | 2016 | [80] | | | | | | Perceived organizational support | + | | Thriving |
| Xu | 2017 | [88] | | | | | | Psychological safety | + | | Thriving |

**Table A3.** *Cont.*

| | Year | Source | Independent Variable (IV) | | | | | | Relationship between IV and DV | Moderator | Dependent Variable (DV) |
|---|---|---|---|---|---|---|---|---|---|---|---|
| | | | Competence | Behavior | Psychological State | Personality | Supervisor | Environment | | | |
| Mushtaq | 2017 | [180] | | | | | | Workplace civility | + | | Thriving |
| Franco-Santos | 2017 | [154] | | | | | | Collegial governance practices of people in academic roles (without leadership re-sponsibilities) | + | | Thriving |
| Franco-Santos | 2017 | [154] | | | | | | Collegial governance practices of people in academic roles (leadership re-sponsibilities) | + | | Thriving |
| Franco-Santos | 2017 | [154] | | | | | | Control governance practices of people in academic roles (leadership re-sponsibilities) | + | | Thriving |
| Franco-Santos | 2017 | [154] | | | | | | Collegial governance practices of people in professional services roles (without leadership re-sponsibilities) | + | | Thriving |
| Franco-Santos | 2017 | [154] | | | | | | Collegial governance practices of people in professional services roles (leadership re-sponsibilities) | + | | Thriving |
| Franco-Santos | 2017 | [154] | | | | | | Control governance practices of people in professional services roles (without leadership re-sponsibilities) | + | | Thriving |
| Bensemmane | 2018 | [135] | | | | | | Transient overall team justice | + | | Thriving |
| Els | 2018 | [309] | | | | | | Perceived or-ganizational support for strengths use | + | | Thriving |
| Els | 2018 | [309] | | | | | | Perceived or-ganizational support for deficit improvement | + | | Thriving |
| Taneva | 2018 | [238] | | | | | | Human resource practices | + | | Thriving |
| Gabriel | 2018 | [68] | | | | | | Female-instigated incivility | - | | Thriving |
| Zhao | 2018 | [266] | | | | | | Workplace violence | - | | Thriving |
| Frazier | 2018 | [310] | | | | | | Employee psychological safety | + | | Thriving |
| Abid | 2018 | [267] | | | | | | Civility | + | | Thriving |
| Silen | 2019 | [292] | | | | | | Person centered climate | + | | Thriving |
| Mahomed | 2019 | [311] | | | | | | Perceived or-ganizational support for strength use | + | | Thriving |
| Sun | 2019 | [312] | | | | | | Mutual under-standing | + | | Thriving |
| Sun | 2019 | [312] | | | | | | Reciprocal favor | + | | Thriving |
| Sun | 2019 | [312] | | | | | | Relationship harmony | + | | Thriving |
| Zhang | 2019 | [313] | | | | | | High-performance work system | + | Proactive personality (-) | Thriving |
| Elahi | 2019 | [314] | | | | | | Colleagues show compassion | + | | Thriving |
| Elahi | 2019 | [314] | | | | | | Colleagues show civility | + | | Thriving |
| Jiang | 2019 | [315] | | | | | | Psychological safety | + | | Thriving |
| Kaltenbrunner | 2019 | [316] | | | | | | Lean maturity | + | | Thriving |
| Kaltenbrunner | 2019 | [316] | | | | | | Job resources | + | | Thriving |
| Paleari | 2019 | [254] | | | | | | Quality of intergroup contact | + | | Thriving |
| Guo | 2019 | [317] | | | | | | Team empow-erment climate | + | | Thriving |

**Table A3.** *Cont.*

| | Year | Source | Independent Variable (IV) | | | | | Relationship between IV and DV Environment | | Moderator | Dependent Variable (DV) |
|---|---|---|---|---|---|---|---|---|---|---|---|
| | | | Competence | Behavior | Psychological State | Personality | Supervisor | | | | |
| Guo | 2019 | [317] | | | | | | Team innovation climate | + | | Thriving |
| Gibson | 2020 | [123] | | | | | | Intrapersonal identity conflict among identities in the workplace | - | | Thriving |
| Chang | 2020 | [318] | | | | | | Psychological contract fulfillment | + | | Thriving |
| Chang | 2020 | [318] | | | | | | Perceived organizational support | + | | Thriving |
| Dimitrova | 2020 | [319] | | | | | | International business travel with high responsibility for people | + | Responsibility for people (+) | Thriving |
| Dimitrova | 2020 | [319] | | | | | | International business travel with low responsibility for people | - | Responsibility for people (+) | Thriving |
| Zeng | 2020 | [296] | | | | | | Psychological safety | + | | Thriving |
| Zhai | 2020 | [297] | | | | | | Co-worker support | + | | Thriving |
| Riaz | 2020 | [257] | | | | | | Strong relational ties with colleagues | + | | Thriving |
| Riaz | 2020 | [257] | | | | | | Weak relational ties with colleagues | + | | Thriving |
| Mahomed | 2020 | [242] | | | | | | Training and development | + | | Thriving |
| Aeschbach | 2020 | [110] | | | | | | Mindfulness intervention | + | | Thriving |
| Qui | 2020 | [320] | | | | | | Citizenship fatigue | - | | Thriving |
| Boyd | 2020 | [321] | | | | | | Sense of community | + | | Thriving |
| Jo | 2020 | [298] | | | | | | Experienced service-oriented high-performance work systems (HPWS) | + | | Thriving |
| Xu | 2020 | [322] | | | | | | Team–Member Exchange (TMX) | + | | Thriving |
| Boyd | 2020 | [323] | | | | | | Sense of community | + | | Thriving |
| Abid | 2020 | [271] | | | | | | Workplace civility | + | | Thriving |
| Mansur | 2020 | [244] | | | | | | Positive career shocks | + | | Thriving |
| Cheng | 2021 | [95] | | | | | | Problems at home | - | Home–work segmentation (-) | Thriving |
| Guan | 2020 | [272] | | | | | | Needs-supplies fit | + | | Thriving |
| Nawaz | 2020 | [181] | | | | | | Workplace incivility | - | Psychological capital (-) | Thriving |
| Sahin | 2021 | [305] | | | | | | Work-to-family conflict | - | | Thriving |
| Farid | 2021 | [324] | | | | | | Organizational justice | + | Servant leadership (+) | Thriving |
| Ren | 2021 | [325] | | | | | | Family–work enrichment | + | | Thriving |
| Ren | 2021 | [325] | | | | | | Family–work conflict | - | | Thriving |
| Pace | 2021 | [275] | | | | | | HRM practices perception | + | | Thriving |
| Kunzelmann | 2021 | [326] | | | | | | Job complexity | + | | Thriving |
| Oliveira | 2021 | [184] | | | | | | Age-inclusive HR practices | + | | Thriving |
| Zhu | 2021 | [277] | | | | | | Workplace violence | - | | Thriving |
| Zhu | 2021 | [277] | | | | | | Perceived organizational support | + | | Thriving |
| Zhu | 2021 | [277] | | | | | | Workplace mindfulness | + | | Thriving |
| Zhu | 2021 | [277] | | | | | | Organizational justice | + | | Thriving |
| Yan | 2021 | [327] | | | | | | Corporate social responsibility | + | Task significance (-) | Thriving |

**Table A3.** *Cont.*

| | Year | Source | Independent Variable (IV) | | | | | Relationship between IV and DV Environment | Moderator | Dependent Variable (DV) |
|---|---|---|---|---|---|---|---|---|---|---|
| | | | Competence | Behavior | Psychological State | Personality | Supervisor | | | |
| Yang | 2021 | [251] | | | | | | Work demands hindrance stressors | - | Thriving |
| Yang | 2021 | [251] | | | | | | Work resources challenge stressors | + | Thriving |
| Sheng | 2021 | [328] | | | | | | Decent work (eg. treated with dignity) | + | Thriving |
| Ozcan | 2021 | [276] | | | | | | Mindfulness-Based Thriving Program | + | Thriving |
| Babalola | 2020 | [329] | | | | | | Perceived Competitive Climate | + | Employee trait competitiveness (+) | Thriving |
| Yi-Feng Chen | 2021 | [330] | | | | | | Perceived organizational support for strengths use | + | Thriving |

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
