# Peer review of "Personal Energy at Work: A Systematic Review"

_sustainability, doi:10.3390/su132313490_

Round 1

Reviewer 1 Report

The paper is interesting and it contributes to update readers in the field of HR. It has several theoretical and practical implications in relation to personal energy at work. It also suggests some limitations and further research. However, it is recommended minor spell check.

Author Response

Thank you for your positive and constructive comments. We have executed an external English spelling check. The changes can be found in the document with track changes.

Reviewer 2 Report

This review paper shall give more visual figure or chart to indicate the next research recommendation for our readers. 

Author Response

Thank you for your constructive feedback. We have added a table (Table 4) that summarizes future research directions. In addition, we have executed an external English spelling check. The changes can be found in the document with track changes.

Reviewer 3 Report

Good day, dear ! 324 references ? 79 orinality?  - to 95 ! 

Author Response

Thank you for your feedback. In an attempt to decrease the number of references we have agreed with Mrs. Nevena Brdarić to place the references of the appendices tables in the appendices. The reason is that our review study contains the evaluation of about 300 papers based on the PRISMA method, therefore it is not possible to reduce references of the total study. With this solution have made the main paper shorter in length with only 128 references.

Thank you for uploading the overview of originality. We concluded that most duplications were found in:

  • the references
  • they derived from ordinary sentences like ‘in the theoretical framework …’
  • cited survey items
  • names of constructs mentioned in literature and therefore exactly copied in the review tables
  • a couple of sentences were similar as used in other literature, we have made changes accordingly by using track changes

We hope that our adjustments are a proper respond to your comments and serve the manuscript well.